# Learning Utilities and Equilibria in Non-Truthful Auctions

**Hu Fu**
Department of Computer Science
University of British Columbia
Vancouver, BC V6T 1Z4
hufu@cs.ubc.ca

**Tao Lin**[*]
Center on Frontiers of Computing Studies
Department of Computer Science
Peking University
Beijing, China
lin_tao@pku.edu.cn

## Abstract

In non-truthful auctions, agents' utility for a strategy depends on the strategies of the opponents and also the prior distribution over their private types; the set of Bayes Nash equilibria generally has an intricate dependence on the prior. Using the First Price Auction as our main demonstrating example, we show that $\tilde{O}(n/\epsilon^2)$ samples from the prior with $n$ agents suffice for an algorithm to learn the interim utilities for all monotone bidding strategies. As a consequence, this number of samples suffice for learning all approximate equilibria. We give almost matching (up to polylog factors) lower bound on the sample complexity for learning utilities. We also consider a setting where agents must pay a search cost to discover their own types. Drawing on a connection between this setting and the first price auction, discovered recently by Kleinberg et al. (2016), we show that $\tilde{O}(n/\epsilon^2)$ samples suffice for utilities and equilibria to be estimated in a near welfare-optimal descending auction in this setting. En route, we improve the sample complexity bound, recently obtained by Guo et al. (2019), for the Pandora's Box problem, which is a classical model for sequential consumer search.

## 1 Introduction

Mechanism design devises systems in which multiple agents take strategic actions based on their private preferences (designated as *types*). For example, an auctioneer devises rules that determine an auction's winner and payments, based on bidders' actions (the bids); the bidders then, knowing the rule and their types — in this case their own values for the item at sale — strategize over their bids. The following task is central to many aspects of mechanism design: given agents' strategies, evaluate each agent's performance, or *utility*. To start with, agents are most often interested in predicting the performance of their strategies given what the other agents do; nowadays, auctioneers and third-party service providers often give guidance to bidding, and are interested in such evaluations as well. An auctioneer often would like to find out if a profile of strategies best respond to each other and are hence *at equilibrium*; revenue, welfare and surplus analysis at equilibrium is all based on utility estimation. Recent development in online ad auctions (such as the oCPX auctions) sees growing popularity of delegated bidding, where bidders entrust the auctioneer/platform with the task of bidding. Auctioneers in this scenario must estimate the bidders' utilities given their bidding strategies. As another application of utility estimation, one may be interested in knowing how far a mechanism is from being *incentive compatible*, i.e., supposing all other agents bid their true values, how much incentive an agent would have to lie about her value (see, e.g., Balcan et al., 2019).

---

[*]Tao Lin is now at Harvard University. This work was done when he was at Peking University and during a visit to the University of British Columbia.

Since a bidder's individual type is known only to herself but may greatly affect her actions and the payoff of everyone else — a scenario known as an *incomplete information game* — more is needed for utility prediction. The canonical paradigm assumes that each agent's type is drawn from a distribution (Harsanyi, 1967). The utility prediction problem becomes, supposing agents other than $i$ use given strategies and their types are drawn from the distributions, what is agent $i$'s utility *in expectation* for using a certain strategy? This so-called *interim* utility is at heart of much modern mechanism design literature from both economics and computer science (e.g. Myerson, 1981; Laffont et al., 1995; Syrgkanis and Tardos, 2013; Hartline and Lucier, 2015).

As is natural for many other settings with distributional prior knowledge, it is important to know how sensitive the problem is to the accuracy of the prior distributions; in particular, given sample access to the distribution, how many samples are needed to estimate the interim utilities of agents, for any profile of strategies? This is the question we study in this work.

As we discuss below, without appealing to the structure of a game, it is hopeless to learn all utilities with a finite number of samples. In this work we take the First Price Auction (FPA) and the All Pay auction as two examples of non-truthful auction, and show that a small number of samples from the type distributions suffice to learn interim utilities for all *monotone* bidding strategies.

The first price auction is one of the most ubiquitous and fundamental auctions, with a large literature on its various aspects, and is gaining more popularity recently for various practical reasons (see e.g. Sluis, 2019; Rajan, 2019; Akbarpour and Li, 2018). In the FPA, the item is sold to the bidder with the highest bid, who then pays her own bid; all other bidders pay nothing. Our main result is that, for an FPA with $n$ bidders, $\tilde{O}(n/\epsilon^2)$ samples from the type distribution suffice to learn, with high probability, the interim utilities up to $\epsilon$ additive error for any profile of monotone bidding strategies.[2] As a corollary, this number of samples also suffice to learn the set of all $\epsilon$-Bayes equilibria of an FPA, and to compute an $\epsilon$-Bayes equilibrium in polynomial time. We also show that our bound is tight up to a logarithmic factor.

**Sampling from Type Distributions.** The assumption of having sample access to the underlying type distribution is standard in a long line of work on learning in mechanism design (e.g. Cole and Roughgarden, 2014; Morgenstern and Roughgarden, 2015, 2016; Balcan et al., 2016, 2018; Gonczarowski and Nisan, 2017; Syrgkanis, 2017; Gonczarowski and Weinberg, 2018; Guo et al., 2019b). In particular, type samples have been assumed in the context of learning in non-truthful auctions (Balcan et al., 2019). Just as in classical microeconomics, prior knowledge (in the form of samples here) comes from market research, survey, simulation etc., and is not assumed to be from past bidding history. We distance our approach from the line of work on learning non-truthful auctions where samples are from past bidding history (Chawla et al., 2017; Hartline and Taggart, 2019). This latter approach, with obvious merits, has its limitations. Crucially, it assumes that the observed bidding in a non-truthful mechanism is at equilibrium, which may not be the case in reality; also, to avoid strategic issues between auctions, the bidders need to be short lived or myopic strategically. The two approaches (type samples vs. bid samples) complement each other even in learning non-truthful mechanisms; this work takes the first approach, and leaves the direction with bid samples as an enticing open question.

**Our Techniques.** Let us draw an analogy with classical PAC learning, where one typically has a class of functions on an input space and an unknown distribution over the input; with samples from the input distribution, one is to learn the expectation of the functions on the unknown distribution. In our setting, every profile of strategies, one for each agent, defines a function which maps a profile of types to a utility for each agent; given the underlying value distribution, the expectation of this function is the agents' interim utilities. We would like to have a number of samples such that, with high probability, the empirical estimation of utilities for *all* bidding strategies is approximately correct.

With any finite number of samples, it is impossible to learn utilities for all arbitrary bidding strategies. Consider the following example: in an auction with two bidders, the first bidder having value 1 and bidding $\frac{1}{2}$, and the second bidder's value $v_2$ uniformly drawn from $[0, 1]$; any finite set of samples of $v_2$ has probability measure 0 in the distribution of $v_2$; therefore on any set of samples, there are bidding strategies of bidder 2 that look the same on the sampled values but give bidder 1 drastically different utilities in expectation on the value distribution.

To overcome this, we observe that non-monotone bidding strategies are (weakly) dominated by monotone ones in a first price auction — a bidding strategy is monotone if its bid is non-decreasing with the value. For all practical purposes, therefore, learning utilities for monotone bidding strategies suffices. A key technical insight of this work is that the restriction to monotone strategies makes it possible to learn utilities with a small number of samples.

Our sample complexity bound comes from bounding the *pseudo-dimension* of the class of utility functions for monotone bidding strategies. Pseudo-dimension is a generalization of the VC-dimension, and has been applied to learning in mechanism design (Morgenstern and Roughgarden, 2015, 2016; Balcan et al., 2018, 2019). Another key step is a lemma (Lemma 3.10), highly related to a concentration inequality shown by Devanur et al. (2016), that translates learning on a correlated distribution from samples to learning on a natural empirical product distribution.

We extend our main result to the *all pay auction*. In the all pay auction, the item is sold to the highest bidder, but all bidders, including the losers, pay their bids. This auction is a good model for crowdsourcing (Chawla et al., 2012).

**Lower Bound.** We give a nontrivial, information-theoretic lower bound for the utility learning problem, showing that $\Omega(n/\epsilon^2)$ samples are needed for learning utilities with monotone bidding strategies. The polynomial dependence on $n$, the number of bidders, is interesting. In particular, this highlights the difference between learning utilities for a fixed profile of bidding strategies and our task, which is to simultaneously learn the utilities for all strategies. For the first task, one may simply estimate, for each bidder, the cumulative density function of the highest bid among the opponents; it takes only $\Theta(1/\epsilon^2)$ samples for the estimated CDF to be accurate everywhere.

**Auctions with Search Costs.** We extend our sample complexity bound to auctions with *search costs*. In this setting, bidders know their value distributions but need to incur a cost to learn their own values. Real estate and insurance markets exhibit this feature. Kleinberg et al. (2016) showed that the Dutch auction beats the English auction and is near optimal at equilibrium for welfare in this setting. If the value distributions are unknown, how many samples are needed for the Dutch auction on the empirical distribution to be near optimal? A real estate market maker may be interested in this: he may collect market information and suggest search and bidding strategies for the participants. We show that $\tilde{O}(n/\epsilon^2)$ samples again suffice.

En route, we improve a sample complexity bound, recently obtained by Guo et al. (2019a), for the *Pandora's Box* problem (Weitzman, 1979), a classical model and algorithm for sequential search.

**Additional Related Works** Most works cited above on learning in mechanisms are on learning revenue optimal mechanisms. The following two exceptions are particularly close to our work. Given a non-truthful auction, Balcan et al. (2019) studied the number of value samples needed to learn the maximal interim utility a bidder with some value could gain by non-truthful bidding, when all other bidders are truthful. In comparison, we learn interim utilities when all bidders use arbitrary monotone bidding strategies; this suffices for the study of virtually all properties of an auction, including the task of Balcan et al. (2019) [3]. Our results rely on the monotonicity of the bidding strategy, which is natural in single item auctions; whether extension is possible to multi-parameter settings in a similar manner is an interesting question.

Areyan Viqueira et al. (2019) studied learning utilities in simulation-based games. They considered abstract conditional normal-form games, where the conditions play the role of types in auctions. They used Rademacher complexity, similar to pseudo-dimension, to bound the number of sample conditions, but did not give such bounds for concrete games. Bounding these complexity measures is generally challenging if not impossible. For example, without monotonicity in bidding strategies, the pseudo-dimension of utilities even in first price auctions is unbounded, as we discussed above.

There is a large literature on the computation of equilibrium in the first price auction (e.g. Marshall et al., 1994; Gayle and Richard, 2008; Escamocher et al., 2009). There has been major recent progress

on the problem for discrete distributions (Shen et al., 2020). Our work provides additional motivation to study the problem on discrete distributions.

Guo et al. (2019a) gave a general method to bound sample complexity for learning over product distributions. Their main result requires a "strong monotonicity" property on the problem, which is satisfied by the Pandora Box problem. Utilities for monotone bidding strategies, however, do not satisfy their property; our bound for the Pandora Box problem also slightly improves over theirs.

## 2 Preliminaries on Auctions

**Auctions.** In a single item auction, $n$ bidders compete for the item. We use $[n]$ to denote $\{1, \ldots, n\}$. Each bidder $i \in [n]$ has a private value $v_i$ drawn from a distribution $F_i$ supported on $T_i \subseteq \mathbb{R}_+$. $|T_i|$ can be infinite. Distributions of different bidders are independent and can be non-identical. In the auction, each bidder $i$ makes a sealed-envelope bid of $b_i \geq 0$, and the auction maps the vector of bids $\boldsymbol{b}$ to an *allocation* and *payments*. $x_i(\boldsymbol{b}) \in [0, 1]$ denotes the probability with which bidder $i$ receives the item at bid vector $\boldsymbol{b}$, with $\sum_{i=1}^n x_i(\boldsymbol{b}) \leq 1$, and $p_i(\boldsymbol{b})$ is the (expected) payment made by bidder $i$. In the *first price auction* (FPA), the highest bidder (assuming there is no tie) wins the item and pays her bid, and no other bidder pays anything. In the *all pay auction*, the highest bidder wins the item but every bidder pays her bid.

Bidder $i$'s *ex post utility* is $U_i(v_i, b_i, \boldsymbol{b}_{-i}) := v_i x_i(\boldsymbol{b}) - p_i(\boldsymbol{b})$.

As is standard in auction theory, we use $\boldsymbol{v}_{-i}$ to denote the vector $(v_1, \ldots, v_{i-1}, v_{i+1}, \ldots, v_n)$. Other vectors with subscript "$-i$" are similarly defined.

**Strategies and Equilibrium.** A *bidding strategy* maps a bidder's value to a bid. With slight abuse of notation, we denote bidder $i$'s strategy as $b_i : T_i \rightarrow \mathbb{R}_+$. To distinguish a bidding strategy (a mapping) and a bid (a number), we write the former as $b_i(\cdot)$ and the latter as $b_i$. We write a vector of bids $(b_1(v_1), \cdots, b_n(v_n))$ as $\boldsymbol{b}(\boldsymbol{v})$, and denote by $\boldsymbol{b}_{-i}(\boldsymbol{v}_{-i})$ the bids made by bidders except bidder $i$.

Bidder $i$ with value $v_i$ and bidding $b_i$, while the other bidders use bidding strategies $\boldsymbol{b}_{-i}(\cdot)$, has *interim utility*

$$u_i(v_i, b_i, \boldsymbol{b}_{-i}(\cdot)) := \mathbf{E}_{\boldsymbol{v}_{-i} \sim \boldsymbol{F}_{-i}}\left[U_i(v_i, b_i, \boldsymbol{b}_{-i}(\boldsymbol{v}_{-i}))\right] \tag{1}$$
$$= \mathbf{E}_{\boldsymbol{v}_{-i} \sim \boldsymbol{F}_{-i}}\left[v_i x_i(b_i, \boldsymbol{b}_{-i}(\boldsymbol{v}_{-i})) - p_i(b_i, \boldsymbol{b}_{-i}(\boldsymbol{v}_{-i}))\right].$$

**Definition 2.1.** *For $\epsilon \geq 0$, a profile of bidding strategies $\boldsymbol{b}(\cdot) = (b_1(\cdot), \cdots, b_n(\cdot))$ in an auction is an $\epsilon$-Bayes Nash equilibrium ($\epsilon$-BNE) for value distribution $\boldsymbol{F} = \prod_{i=1}^n F_i$ if for each bidder $i \in [n]$ and each $v_i \in T_i$, for any $b_i' \in \mathbb{R}_+$, $u_i(v_i, b_i(v_i), \boldsymbol{b}_{-i}(\cdot)) \geq u_i(v_i, b_i', \boldsymbol{b}_{-i}(\cdot)) - \epsilon$. If $\epsilon = 0$, $\boldsymbol{b}(\cdot)$ is a Bayes Nash equilibrium.*

A bidding strategy $b_i(\cdot)$ is said to be *monotone* if $v \geq v' \Rightarrow b_i(v) \geq b_i(v')$.

**Proposition 2.2.** *In a first price auction or an all pay auction, for any bidding strategy $b_i(\cdot)$ of bidder $i$, for any value distributions $\boldsymbol{F}_{-i}$, and any bidding strategies $\boldsymbol{b}_{-i}(\cdot)$, there is a monotone bidding strategy $b_i'(\cdot)$ such that $\forall v_i \in T_i$, $u_i(v_i, b_i'(v_i), \boldsymbol{b}_{-i}(\cdot)) \geq u_i(v_i, b_i(v_i), \boldsymbol{b}_{-i}(\cdot))$.*

The proof is in the supplementary file.

Throughout the paper we assume that there is an upper bound $H$ on $T_i$. Therefore, no bidder is willing to bid $b_i > H$ and we only consider bidding strategies $b_i(\cdot)$ that map $T_i$ to $[0, H]$.

## 3 Sample Complexity of Utility Estimation

We are given a set of $m$ samples, $\boldsymbol{s} = (\boldsymbol{s}^1, \ldots, \boldsymbol{s}^m)$ from the value distribution $\boldsymbol{F} = \prod_i F_i$. Note that each sample $\boldsymbol{s}^j$ is a vector of values $(s_1^j, \ldots, s_n^j)$ drawn from $\boldsymbol{F}$.

Given value samples $\boldsymbol{s}$, a *utility learning algorithm* $\mathcal{A}$, for each bidder $i$ with value $v_i$ and bidding strategies $\boldsymbol{b}(\cdot) = (b_1(\cdot), \cdots, b_n(\cdot))$, outputs $\mathcal{A}(\boldsymbol{s}, i, v_i, \boldsymbol{b}(\cdot))$, which estimates bidder $i$'s interim utility when her value is $v_i$ and the bidders use bidding strategies $\boldsymbol{b}(\cdot)$.

**Definition 3.1.** *Let $\mathcal{B}$ be a set of bidding strategy profiles. For $\epsilon > 0, \delta \in (0, 1)$, a utility learning algorithm $\mathcal{A}$ $(\epsilon, \delta)$-learns with $m$ samples the utilities over $\mathcal{B}$ if, for any product value distribution $\boldsymbol{F}$,*

*with probability at least $1 - \delta$, for any bidding strategy profile $\boldsymbol{b}(\cdot) \in \mathcal{B}$, for each bidder $i \in [n]$ and any $v_i \in T_i$,*

$$|\mathcal{A}(\boldsymbol{s}, i, v_i, \boldsymbol{b}(\cdot)) - u_i(v_i, b_i(v_i), \boldsymbol{b}_{-i}(\cdot))| < \epsilon,$$

*where the randomness is over the random draw of samples and the randomness of $\mathcal{A}$ if it is randomized.*

Throughout the paper we will take $\mathcal{B}$ to be the set of profiles of monotone bidding strategies (see Proposition 2.2).

## 3.1 Upper Bound on Sample Complexity

In this section, we show that $\tilde{O}(n/\epsilon^2)$ value samples suffice for learning the interim utilities for all monotone bidding strategies. The learning algorithm is the empirical distribution estimator, which outputs expected utilities on the uniform distribution over the samples.

**Definition 3.2.** *The* empirical distribution estimator, *denoted by* $\mathrm{Emp}$, *estimates interim utilities on the uniform distribution over the samples. Formally, on samples $\boldsymbol{s} = (\boldsymbol{s}^1, \ldots, \boldsymbol{s}^m)$, for bidder $i$ with value $v_i$, for bidding strategies $\boldsymbol{b}(\cdot)$,*

$$\mathrm{Emp}(\boldsymbol{s}, i, v_i, \boldsymbol{b}(\cdot)) \coloneqq \frac{1}{m} \sum_{j=1}^{m} U_i(v_i, b_i(v_i), \boldsymbol{b}_{-i}(\boldsymbol{s}_{-i}^j)).$$

One more technicality concerns what happens when more than one bidder makes the highest bid. Even though in any equilibrium of a first price auction, this must happen with probability $0$, for the utility learning problem to be well defined we need to specify tie-breaking rules. As we shall see, the tie-breaking rule does not affect the sample complexity. We consider here two tie-breaking rules: by *random-allocation* rule, the item is assigned to one of the highest bidders uniformly at random; by *no-allocation* rule, no one wins the item whenever there is a tie for the highest bid.[4]

**Theorem 3.3.** *Suppose $T_i \subseteq [0, H]$ for each $i \in [n]$, and the tie-breaking rule is random-allocation or no-allocation. For any $\epsilon > 0, \delta \in (0, 1)$, there is*

$$M = O\left(\frac{H^2}{\epsilon^2}\left[n \log n \log\left(\frac{H}{\epsilon}\right) + \log\left(\frac{n}{\delta}\right)\right]\right), \tag{2}$$

*such that for any $m \geq M$, the empirical distribution estimator $\mathrm{Emp}$ $(\epsilon, \delta)$-learns with $m$ samples the utilities over the set of all monotone bidding strategies.*

### 3.1.1 Pseudo-dimension and the Proof of Theorem 3.3

Pseudo-dimension is a well known tool for upper bounding sample complexity (see, e.g. Anthony and Bartlett, 2009), and has been applied to learning in mechanism design (Morgenstern and Roughgarden, 2015, 2016; Balcan et al., 2018, 2019).

**Definition 3.4.** *Given a class $\mathcal{H}$ of real-valued functions on input space $\mathcal{X}$, a set of input $x_1, \ldots, x_m$ is said to be* pseudo-shattered *if there exist* witnesses *$t_1, \ldots, t_m \in \mathbb{R}$ such that for any label vector $\boldsymbol{l} \in \{1, -1\}^m$, there exists $h_{\boldsymbol{l}} \in \mathcal{H}$ such that $\mathrm{sgn}(h_{\boldsymbol{l}}(x_i) - t_i) = l_i$ for each $i = 1, \ldots, m$, where $\mathrm{sgn}(y) = 1$ if $y > 0$ and $-1$ if $y < 0$. The* pseudo-dimension *of $\mathcal{H}$, $\mathrm{Pdim}(\mathcal{H})$, is the size of the largest set of inputs that can be pseudo-shattered by $\mathcal{H}$.*

**Definition 3.5.** *For $\epsilon > 0, \delta \in (0, 1)$, a class of functions $\mathcal{H} : \mathcal{X} \to \mathbb{R}$ is $(\epsilon, \delta)$-uniformly convergent with sample complexity $M$ if for any $m \geq M$, for any distribution $F$ on $\mathcal{X}$, if $s^1, \ldots, s^m$ are i.i.d. samples from $F$, with probability at least $1 - \delta$, for every $h \in \mathcal{H}$, $\left|\mathbf{E}_{x \sim F}[h(x)] - \frac{1}{m} \sum_{j=1}^{m} h(s^j)\right| < \epsilon$.*

**Theorem 3.6** (See Anthony and Bartlett, 2009). *Let $\mathcal{H}$ be a class of functions with range $[0, H]$ and pseudo-dimension $d = \mathrm{Pdim}(\mathcal{H})$, for any $\epsilon > 0$, $\delta \in (0, 1)$, $\mathcal{H}$ is $(\epsilon, \delta)$-uniformly convergent with sample complexity $O\left(\left(\frac{H}{\epsilon}\right)^2 [d \log(\frac{H}{\epsilon}) + \log(\frac{1}{\delta})]\right)$.*

We show Theorem 3.3 by treating the utilities on monotone bidding strategies as a class of functions, whose uniform convergence implies that Emp learns the interim utilities.

For each bidder $i$, let $h^{v_i, \boldsymbol{b}(\cdot)}$ be the function that maps the opponents' values to bidder $i$'s ex post utility, that is,

$$h^{v_i, \boldsymbol{b}(\cdot)}(\boldsymbol{v}_{-i}) = U_i(v_i, b_i(v_i), \boldsymbol{b}_{-i}(\boldsymbol{v}_{-i})).$$

Let $\mathcal{H}_i$ be the set of all such functions corresponding to the set of monotone strategies,

$$\mathcal{H}_i = \left\{ h^{v_i, \boldsymbol{b}(\cdot)}(\cdot) \mid v_i \in T_i, \ \boldsymbol{b}(\cdot) \text{ is monotone} \right\}.$$

By (1), the expectation of $h^{v_i, \boldsymbol{b}(\cdot)}(\cdot)$ over $\boldsymbol{F}_{-i}$ is the interim utility of bidder $i$:

$$\mathbf{E}_{\boldsymbol{v}_{-i} \sim \boldsymbol{F}_{-i}} \left[ h^{v_i, \boldsymbol{b}(\cdot)}(\boldsymbol{v}_{-i}) \right] = u_i(v_i, b_i(v_i), \boldsymbol{b}_{-i}(\cdot)).$$

By Definition 3.2, on samples $\boldsymbol{s} = (\boldsymbol{s}^1, \dots, \boldsymbol{s}^m)$,

$$\text{Emp}(\boldsymbol{s}, i, v_i, \boldsymbol{b}(\cdot)) = \frac{1}{m} \sum_{j=1}^{m} h^{v_i, \boldsymbol{b}(\cdot)}(\boldsymbol{s}_{-i}^j).$$

Thus,

$$|\text{Emp}(\boldsymbol{s}, i, v_i, \boldsymbol{b}(\cdot)) - u_i(v_i, b_i(v_i), \boldsymbol{b}_{-i}(\cdot))| = \left| \mathbf{E}_{\boldsymbol{v}_{-i}} \left[ h^{v_i, \boldsymbol{b}(\cdot)}(\boldsymbol{v}_{-i}) \right] - \frac{1}{m} \sum_{j=1}^{m} h^{v_i, \boldsymbol{b}(\cdot)}(\boldsymbol{s}_{-i}^j) \right|.$$

$$(3)$$

The right hand side of (3) is the difference between the expectation of $h^{v_i, \boldsymbol{b}(\cdot)}$ on the distribution $\boldsymbol{F}_{-i}$ and that on the empirical distribution with samples drawn from $\boldsymbol{F}_{-i}$. Now by Theorem 3.6, to bound the number of samples needed by Emp to $(\epsilon, \delta)$-learn the utilities over monotone strategies, it suffices to bound the pseudo-dimension of $\mathcal{H}_i$. With the following key lemma, the proof is completed by observing that the range of each $h^{v_i, \boldsymbol{b}(\cdot)}$ is within $[-H, H]$ and by taking a union bound over $i \in [n]$.

**Lemma 3.7.** *If tie breaking is random-allocation or no-allocation, then* $\text{Pdim}(\mathcal{H}_i) = O(n \log n)$.

The proof of Lemma 3.7 follows a powerful framework introduced by Morgenstern and Roughgarden (2016) and Balcan et al. (2018) for bounding the pseudo-dimension of a class $\mathcal{H}$ of functions: given samples that are to be pseudo-shattered and for any (fixed) witnesses, one classifies the functions in $\mathcal{H}$ into categories, so that functions in the same category must output the same label on all the samples; by counting and bounding the number of such categories, one can bound the number of shattered samples. Our proof follows this strategy. To bound the number of categories, we make use of monotonicity of bidding functions, which is specific to our problem.

We give a proof below for the simplest case with two bidders and no-allocation tie-breaking rule, and relegate the full proof to the supplementary file.

*Proof of Lemma 3.7 for a special case.* Consider $n = 2$ and no-allocation tie-breaking rule. Fix an arbitrary set of $m$ samples $\boldsymbol{s}_{-i}^1, \dots, \boldsymbol{s}_{-i}^m$. Consider any set of potential witnesses $(t_1, t_2, \dots, t_m)$. Each hypothesis in $\mathcal{H}_i$ then gives every sample $\boldsymbol{s}_{-i}^j$ a label according to the witness $t_j$, giving rise to a label vector in $\{-1, +1\}^m$. We show that $\mathcal{H}_i$ can be divided into $m + 1$ sub-classes $\mathcal{H}_i^0, \dots, \mathcal{H}_i^m$, such that each sub-class $\mathcal{H}_i^k$ generates at most $m + 1$ different label vectors. Thus $\mathcal{H}_i$ generates at most $(m + 1)^2$ label vectors in total. To pseudo-shatter $\boldsymbol{s}_{-i}^1, \dots, \boldsymbol{s}_{-i}^m$, we need $2^m$ different label vectors; therefore $(m + 1)^2 \geq 2^m$, which implies $m = O(1)$.

We now show how $\mathcal{H}_i$ is thus divided. Note that, for $n = 2$, each $\boldsymbol{s}_{-i}^k$ is just a real number and we can sort them; for ease of notation let $x^k$ denote $\boldsymbol{s}_{-i}^k$ for $k = 1, \dots, m$ and suppose $x^1 \leq x^2 \leq \cdots \leq x^m$. We put hypothesis $h^{v_i, \boldsymbol{b}(\cdot)}$ into the $k$-th sub-class, $\mathcal{H}_i^k$, if

$$b_{-i}(x^k) < b_i(v_i) \ \text{and} \ b_{-i}(x^{k+1}) \geq b_i(v_i).$$

This is well defined because, by assumption, $b_{-i}(x)$ is monotone non-decreasing in $x$.

We now show that each sub-class $\mathcal{H}_i^k$ gives rise to at most $m + 1$ label vectors. For any $h^{v_i, \boldsymbol{b}(\cdot)} \in \mathcal{H}_i^k$, we have $h^{v_i, \boldsymbol{b}(\cdot)}(x^j) = v_i - b_i(v_i)$ for any $j \leq k$ (because bidder $i$'s bid $b_i(v_i)$ is higher than the opponent's), and $h^{v_i, \boldsymbol{b}(\cdot)}(x^j) = 0$ for any $j > k$. On the first $k$ samples $x^1, \ldots, x^k$, any fixed hypothesis $h^{v_i, \boldsymbol{b}(\cdot)}(\cdot) \in \mathcal{H}_i^k$ outputs a constant $v_i - b_i(v_i)$; as one varies this constant and compares it with the $k$ witnesses $t_1, \ldots, t_k$, there are only $k + 1$ possible results from the comparisons. On the remaining $m - k$ samples, only one pattern is possible, since all hypotheses in $\mathcal{H}_i^k$ output 0 on these samples. Therefore, at most $k + 1 \leq m + 1$ label vectors can be generated by $\mathcal{H}_i^k$. □

### 3.1.2 Learning on Empirical Product Distributions and Equilibrium Preservation

The empirical distribution estimator approximates interim utilities with high probability, but this does not immediately imply that one may take the first price auction on the empirical distribution as a close approximation to the auction on the original distribution. This is because the empirical distribution over samples is *correlated* — the values $s_1^j, \ldots, s_n^j$ are drawn as a vector, instead of independently. Standard notions, such as Bayes Nash equilibria, defined on product distributions become intricate on correlated distributions, and there is no reason to expect the latter to correspond to the equilibria in the original auction. Therefore, it is desirable that utilities can also be learned on a *product* distribution arising from the samples, where each bidder's value is independently drawn, uniformly from the $m$ samples of her value. We show that this can indeed be done, without substantial increase in the number of samples. The key technical step, Lemma 3.10, is a reduction from learning on empirical distribution to learning on empirical product distribution. We believe this lemma is of independent interest. In fact, in Section 4 we invoke Lemma 3.10 in a different context, that of learning in Pandora's Box problem; the reduction is crucial there for obtaining a polynomial-time learning algorithm.

**Definition 3.8.** *Given samples $\boldsymbol{s} = (\boldsymbol{s}^1, \ldots, \boldsymbol{s}^m)$, $E_i$ is defined to be the uniform distribution over $\{s_i^1, \ldots, s_i^m\}$. The* empirical product distribution $\boldsymbol{E}$ *is the product distribution* $\boldsymbol{E} := \prod_{i=1}^n E_i$.

**Definition 3.9.** *For $\epsilon > 0, \delta \in (0, 1)$, a class of functions $\mathcal{H} : \prod_{i=1}^n T_i \to \mathbb{R}$ is $(\epsilon, \delta)$-uniformly convergent on product distribution with sample complexity $M$ if for any $m \geq M$, for any product distribution $\boldsymbol{F}$ on $\prod_{i=1}^n T_i$, if $\boldsymbol{s}^1, \ldots, \boldsymbol{s}^m$ are i.i.d. samples from $\boldsymbol{F}$, with probability at least $1 - \delta$, for every $h \in \mathcal{H}$,*

$$|\mathbf{E}_{\boldsymbol{t} \sim \boldsymbol{F}} [h(\boldsymbol{t})] - \mathbf{E}_{\boldsymbol{t} \sim \boldsymbol{E}} [h(\boldsymbol{t})]| < \epsilon,$$

*where $\boldsymbol{E}$ is the empirical product distribution.*

**Lemma 3.10.** *Let $\mathcal{H}$ be a class of functions from a product space $\boldsymbol{T}$ to $[0, H]$. If $\mathcal{H}$ is $(\epsilon, \delta)$-uniformly convergent with sample complexity $m = m(\epsilon, \delta)$, then $\mathcal{H}$ is $\left(2\epsilon, \frac{H\delta}{\epsilon}\right)$-uniformly convergent on product distribution with sample complexity $m$.*

Lemma 3.10 is closely related to a concentration inequality by Devanur et al. (2016). Devanur et al. show that for any single function $h : \boldsymbol{T} \to [0, H]$, the expectation of $h$ on the empirical product distribution is close to its expectation on any product distribution with high probability. Our lemma generalizes this to show a simultaneous concentration for a family of functions, and seems more handy for applications such as ours.

Combining Theorem 3.3 with Lemma 3.10, we derive our learning results on the empirical product distribution.

**Definition 3.11.** *The* empirical product distribution estimator Empp *estimates interim utilities of a bidding strategy on the empirical product distribution $\boldsymbol{E}$. Formally, for bidder $i$ with value $v_i$, for bidding strategy profile $\boldsymbol{b}(\cdot)$,*

$$\text{Empp}(\boldsymbol{s}, i, v_i, \boldsymbol{b}(\cdot)) := \mathbf{E}_{\boldsymbol{v}_{-i} \sim \boldsymbol{E}_{-i}} [U_i(v_i, b_i(v_i), \boldsymbol{b}_{-i}(\boldsymbol{v}_{-i}))]. \tag{4}$$

**Theorem 3.12.** *Suppose $T_i \subseteq [0, H]$ for each $i \in [n]$, and the tie-breaking rule is random-allocation or no-allocation. For any $\epsilon > 0, \delta \in (0, 1)$, there is*

$$M = O\left(\frac{H^2}{\epsilon^2} \left[n \log n \log \left(\frac{H}{\epsilon}\right) + \log \left(\frac{n}{\delta}\right)\right]\right), \tag{5}$$

*such that for any $m \geq M$, the empirical distribution estimator Empp $(\epsilon, \delta)$-learns with $m$ samples the utilities over the set of all monotone bidding strategies.*

By Theorem 3.12, utilities in the FPA on the empirical product distribution approximate those in the FPA on the original distribution, therefore the two auctions share the same set of approximate equilibria:

**Corollary 3.13.** *Suppose $T_i \subseteq [0, H]$ for each $i \in [n]$ and the tie-breaking rule is random-allocation or no-allocation. For any $\epsilon, \epsilon' > 0, \delta \in (0, 1)$, for $m$ satisfying (5), with probability at least $1 - \delta$ over random draws of $s$, for any monotone bidding strategy profile $b(\cdot)$, if $b(\cdot)$ is an $\epsilon'$-BNE in the first price auction on value distribution $E = \prod_i E_i$, then $b(\cdot)$ is an $(\epsilon' + 2\epsilon)$-BNE in the first price auction on value distribution $F = \prod_i F_i$. Conversely, if $b(\cdot)$ is an $\epsilon'$-BNE in the first price auction on value distribution $F$, then $b(\cdot)$ is an $(\epsilon' + 2\epsilon)$-BNE in the first price auction on value distribution $E$.*

Corollary 3.13 has an interesting consequence. Shen et al. (2020) gave a polynomial-time algorithm for computing Bayes Nash equilibrium in first price auctions on discrete value distributions. The empirical product distribution is discrete, so one can run Shen et al.'s algorithm on it. Corollary 3.13 immediately implies:

**Corollary 3.14.** *There is a Monte Carlo randomized algorithm for computing an $\epsilon$-BNE in a first price auction with $n$ bidders on arbitrary product value distributions. The running time of the algorithm is polynomial in $n$ and $\frac{1}{\epsilon}$.*

Note that the running time of the algorithm does not depend on the size of the distributions' support, and works for continuous distributions as well.

Results very similar to Theorem 3.3, Theorem 3.12, and Corollary 3.13, apply to the all pay auction, with the same bounds on the number of samples. The proofs are almost identical and so are omitted.

## 3.2 Lower Bound of Sample Complexity

We give an information theoretic lower bound on the number of samples needed for any algorithm to learn monotone utilities in a first price auction. The lower bound matches our upper bound up to polylog factors.

**Theorem 3.15.** *For any $\epsilon < \frac{1}{4000}, \delta < \frac{1}{20}$, there is a family of product distributions for which no algorithm $(\epsilon, \delta)$-learns, with $m$ samples, utilities over the set of all monotone bidding strategies, for any $m \leq \frac{1}{4 \times 10^8} \cdot \frac{n}{\epsilon^2}$.*

The proof of Theorem 3.15 is in the supplmentary file. As a sketch, the product distributions we construct encode length $n - 1$ binary strings by having slightly unfair Bernoulli distribution for each bidder, the bias shrinking as $n$ grows large. We then show that, if with few samples a learning algorithm learns utilities for all monotone bidding strategies, then there must exist two product distributions from the family which differ at only one coordinate, and yet they can be told apart by the learning algorithm. This must violate well-known information theoretic lower bounds (e.g. Mansour, 2011).

## 4 Auctions with Costly Search

We extend our sample complexity results to auctions in which bidders need to incur a cost to know precisely their values, a model proposed and studied by Kleinberg et al. (2016). Due to page limit, we sketch the results here and relegate most details to the appendix.

In this model, each bidder $i$ knows the distribution $F_i$ from which her value is drawn, but gets to know her value $v_i$ only after incurring a cost $c_i$. This models well, for example, a real estate market, where $c_i$ is an inspection cost. Kleinberg et al. (2016) showed that, due to the search costs, the English auction can have low efficiency, whereas the Dutch auction, with its descending price, can coordinate the bidders' searching in an almost efficient way. Intuitively, a bidder does not inspect her value until the price drops to a certain level, and then either claims the item at the threshold price, or waits till later. In fact, absent incentive issues, this is the procedure by which a central authority would follow to maximize the welfare; the elegant algorithm is known as the Pandora's Box algorithm (Weitzman, 1979). With incentives, bidders shade their bids just as in a first price auction, and there is efficiency loss. This analogy was made precise by Kleinberg et al., who showed a correspondence between the equilibria in a Dutch auction with search costs and the equilibria in a first price auction without search costs but with transformed value distributions. The near efficiency of the Dutch auction therefore

follows from Price of Anarchy results on the first price auction (Syrgkanis and Tardos, 2013; Hoy et al., 2018).

In the appendix we present two sets of results for this setting. We first review the Pandora's Box algorithm, necessary for understanding the correspondence observed by Kleinberg et al. (2016). En route, we show that $\tilde{O}(n/\epsilon^2)$ samples from the value distributions suffice for the algorithm to be $\epsilon$-close to optimal when the distributions are unknown. Our bound slightly improves a recent result by Guo et al. (2019a).

We then review the correspondence between the Dutch auction with search costs and the FPA without search costs. We show that, when the value distributions are unknown, with $\tilde{O}(n/\epsilon^2)$ value samples, all "monotone bidding strategies", when properly defined, in the Dutch auction on the empirical distribution have approximately the same utilities as the corresponding bidding strategies in an FPA defined on a transformation of the original value distribution. As a consequence, the two auctions share the same approximate equilibria, and the Dutch auction at equilibrium on the empirical distribution is nearly efficient.

## 5  Conclusion

In this work we obtained almost tight sample complexity bounds for learning utilities in first price auctions and all pay auctions. Whereas utilities for unconstrained bidding strategies are hard to learn, we show that learning is made possible by focusing on monotone bidding strategies, which is sufficient for all practical purposes. We also extended the results to auctions where search costs are present.

Monotonicity is a natural assumption on bidding strategies in a single item auction, but it does not generalize to multi-parameter settings, where characterization of equilibrium is notoriously difficult. It is an interesting question whether our results can be generalized to multi-item auctions, such as simultaneous first-price auctions, via more general, lossless structural assumptions on the bidding strategies.

Our results also depend on the values being drawn independently. When bidders' values are correlated, the conditional distribution of opponents' values changes with a bidder's value, and any naïve utility learning algorithm needs a number of samples that grows linearly with the size of a bidder's type space. It is interesting whether there are meaningful tractable middle grounds for utility learning between product distributions and arbitrary correlated distributions.

## Broader Impact

This work is theoretical in nature, and should be understood as providing understanding for existing or possible practice, rather than having immediate societal impacts. More accurate utility estimation in auctions can improve market efficiency, improve bidders' profit in the short term, and help maintain the health of the markets in the long term. When such auctions are on online advertisements, for example, users eventually benefit from the long-term health of the markets. Third-party service providers may also help market participants optimize their performances via data collection and effective modeling. The authors do not see other negative ethical aspects or societal consequences, especially given the theoretical nature of the work.

## Acknowledgements

Hu Fu would like to thank Anna Karlin for helpful discussion in early stages of the work. The work was funded by an NSERC Discovery Grant, NSERC Discovery Acceleration Grant, and a Canadian Research Chair stipend.

## Footnotes

[2] The $\tilde{O}(\cdot)$ notation omits polylogarithmic factors.

[3]Our results imply that the maximal interim utility (w.r.t the opponents' value distribution) obtained by non-truthful bidding can be approximated by the maximal obtainable interim utility w.r.t. the empirical distribution, which can be computed by enumerating the samples in the empirical distribution because a best-responding bid must be equal to (or slightly more than) some opponent's value from the empirical distribution.

[4]In fact, our results hold for all tie-breaking rules by which any bidder $i$'s allocation is deterministically determined by the comparisons of her bid $b_i$ with the other bidders' bids, and not affected by any other information (such as the specific value of $b_i$). More formally, the allocation for bidder $i$ is either $0$ or $1$ and is determined by a vector in $\{<, =, >\}^{n-1}$, which records the comparison between $b_i$ and every other bidder's bid. Rules satisfying this condition include no-allocation, breaking ties lexicographically, etc.

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
