[Supplementary Material]

# A  Missing Proofs from Section 2

## A.1  Proof of Proposition 2.2

**Proposition 2.2.** *In a first price auction or an all pay auction, for any bidding strategy $b_i(\cdot)$ of bidder $i$, for any value distributions $\boldsymbol{F}_{-i}$, and any bidding strategies $\boldsymbol{b}_{-i}(\cdot)$, there is a monotone bidding strategy $b_i'(\cdot)$ such that $\forall v_i \in T_i$, $u_i(v_i, b_i'(v_i), \boldsymbol{b}_{-i}(\cdot)) \geq u_i(v_i, b_i(v_i), \boldsymbol{b}_{-i}(\cdot))$.*

*Proof.* For all practical purposes we may assume $b_i(T_i)$ to be compact. Fix the distributions $\boldsymbol{F}_{-i}$ and strategies $\boldsymbol{b}_{-i}(\cdot)$ of other bidders. To simplify notation when $\boldsymbol{b}_{-i}(\cdot)$ is fixed, let the interim allocation $x_i(b_i)$ be $\mathbf{E}_{\boldsymbol{v}_{-i} \sim \boldsymbol{F}_{-i}}[x_i(b_i, \boldsymbol{b}_{-i}(\boldsymbol{v}_{-i}))]$, the interim payment $p_i(b_i) := \mathbf{E}_{\boldsymbol{v}_{-i} \sim \boldsymbol{F}_{-i}}[p_i(b_i, \boldsymbol{b}_{-i}(\boldsymbol{v}_{-i}))]$, and the interim utility $u_i(v_i, b_i) := u_i(v_i, b_i, \boldsymbol{b}_{-i}(\cdot))$. Without loss of generality, we may assume for each $v_i$, $u_i(v_i, b_i(v_i)) = \max_{v \in T_i} u_i(v_i, b_i(v))$ (Otherwise we can first readjust $b_i(\cdot)$ this way, which only weakly improves the utility of all types.)

Suppose $b_i(\cdot)$ is non-monotone, i.e., there exist $v_i' > v_i$, such that $b_i(v_i') < b_i(v_i)$. By the assumption that $u_i(v_i, b_i(v_i)) = \max_{v \in T_i} u_i(v_i, b_i(v))$ for each $v_i$, we have

$$v_i x_i(b_i(v_i)) - p_i(b_i(v_i)) \geq v_i x_i(b_i(v_i')) - p_i(b_i(v_i')); \tag{6}$$

$$v_i' x_i(b_i(v_i')) - p_i(b_i(v_i')) \geq v_i' x_i(b_i(v_i)) - p_i(b_i(v_i)). \tag{7}$$

Adding (6) and (7), we obtain

$$(v_i' - v_i)[x_i(b_i(v_i')) - x_i(b_i(v_i))] \geq 0. \tag{8}$$

Since $v_i' > v_i$, we get

$$x_i(b_i(v_i')) \geq x_i(b_i(v_i)). \tag{9}$$

In both the first price auction and the all pay auction we also have $x_i(b_i(v_i')) \leq x_i(b_i(v_i))$ because the probability that $i$ receives the item cannot decrease if her bid increases. Therefore, it must be

$$x_i(b_i(v_i')) = x_i(b_i(v_i)). \tag{10}$$

Plugging (10) into (6) and (7), we obtain

$$p_i(b_i(v_i')) = p_i(b_i(v_i)). \tag{11}$$

For the all pay auction, since bidder $i$ pays her bid whether or not she wins the item, (11) implies $b_i(v_i) = b_i(v_i')$, a contradiction.

For the first price auction, for any bid $b$ made by bidder $i$, $p_i(b) = b \cdot x_i(b)$. By (11), $b_i(v_i') x_i(b_i(v_i')) = b_i(v_i) x_i(b_i(v_i))$. On the other hand, $x_i(b_i(v_i')) = x_i(b_i(v_i))$ and $b_i(v_i') > b_i(v_i)$, so we must have

$$x_i(b_i(v_i')) = x_i(b_i(v_i)) = 0.$$

In other words, $b_i(v_i)$ must be monotone non-decreasing everywhere except maybe for values whose bids are so low that the bidder does not win and hence obtains zero utility. Letting the bidder bid 0 for all values on which her allocation is 0 does not affect her utility and yields a monotone bidding strategy. $\square$

# B  Missing Proofs from Section 3

## B.1  Upper Bound

### B.1.1  Proof of Lemma 3.7

**Lemma 3.7.** *If tie breaking is random allocation or no allocation, then $\mathrm{Pdim}(\mathcal{H}_i) = O(n \log n)$.*

*Proof.* We discussed the case with $n = 2$ in Section 3.1. Now we consider the general case with $n > 2$ bidders. We give the proof for the random-allocation tie-breaking rule; the proof for the no-allocation rule is similar (and in fact simpler). For ease of notation, we use $\boldsymbol{x}^k$ to denote $\boldsymbol{s}_{-i}^k$. Recall that each $\boldsymbol{x}^k$ is a vector in $\mathbb{R}^{n-1}$. We write its $j$-th component as $x_j^k$. We start with a simple

observation: for any $v_i$ and $\boldsymbol{b}(\cdot)$, the output of $h^{v_i,\boldsymbol{b}(\cdot)}$ on any input $\boldsymbol{x}^k$ must be one of the following $n+1$ values: $v_i - b_i, \frac{v_i - b_i}{2}, \ldots, \frac{v_i - b_i}{n}$, or 0; this value is fully determined by the $n-1$ comparisons $b_i \lesseqgtr b_j(x_j^k)$ for each $j \neq i$. We argue that the hypothesis class $\mathcal{H}_i$ can be divided into $O(m^{2n})$ sub-classes $\{\mathcal{H}_i^{\mathbf{k}}\}_{\mathbf{k} \in [m+1]^{2(n-1)}}$ such that each sub-class $\mathcal{H}_i^{\mathbf{k}}$ generates at most $O(m^n)$ different label vectors. Thus $\mathcal{H}_i$ generates at most $O(m^{3n})$ label vectors in total. To pseudo-shatter $m$ samples, we need $O(m^{3n}) \geq 2^m$, which implies $m = O(n \log n)$.

We now define sub-classes $\{\mathcal{H}_i^{\mathbf{k}}\}_{\mathbf{k}}$, each indexed by $\mathbf{k} \in [m+1]^{2(n-1)}$. For each dimension $j \neq i$, we sort the $m$ samples by their $j$-th coordinates non-decreasingly, and use $\pi(j, \cdot)$ to denote the resulting permutation over $\{1, 2, \ldots, m\}$; formally, let $x_j^{\pi(j,1)} \leq x_j^{\pi(j,2)} \leq \cdots \leq x_j^{\pi(j,m)}$. For each hypothesis $h^{v_i,\boldsymbol{b}(\cdot)}(\cdot)$, for each $j$, we define two special positions; these positions are similar to the position $k$ in the case for two bidders; we now need a pair, because of the need to keep track of ties, due to the more complex random-allocation tie-breaking rule. Let $k_{j,1}$ be $\max\{0, \{k : b_j(x_j^{\pi(j,k)}) < b_i(v_i)\}\}$, and let $k_{j,2}$ be $\min\{m+1, \{k : b_j(x_j^{\pi(j,k)}) > b_i(v_i)\}\}$. As in the case for two bidders, this is well defined because of the monotonicity of $b_j(\cdot)$. It also follows that, if $k_{j,1} < k_{j,2} - 1$, then for any $k$ such that $k_{j,1} < k < k_{j,2}$, we must have $b_j(x_j^{\pi(j,k)}) = b_i(v_i)$. A hypothesis $h^{v_i,\boldsymbol{b}(\cdot)}(\cdot)$ belongs to sub-class $\mathcal{H}_i^{\mathbf{k}}$ where the index $\mathbf{k}$ is $(k_{j,1}, k_{j,2})_{j \in [n] \setminus \{i\}}$. The number of sub-classes is clearly bounded by $(m+1)^{2(n-1)}$.

We now show that the hypotheses within each sub-class $\mathcal{H}_i^{\mathbf{k}}$ give rise to at most $(m+1)^n$ label vectors. Let us focus on one such class with index $\mathbf{k}$. On the $k$-th sample $\boldsymbol{x}^k$, a hypothesis's membership in $\mathcal{H}_i^{\mathbf{k}}$ suffices to specify whether bidder $i$ is a winner on this sample, and, if so, the number of other winning bids at a tie. Therefore, the class index $\mathbf{k}$ determines a mapping $c : [m] \to \{0, 1, \ldots, n\}$, with $c(k) > 0$ meaning bidder $i$ is a winner on sample $\boldsymbol{x}^k$ at a tie with $c(k) - 1$ other bidders, and $c(k) = 0$ meaning bidder $i$ is a loser on sample $\boldsymbol{x}^k$. The output of a hypothesis $h^{v_i,\boldsymbol{b}(\cdot)}(\cdot) \in \mathcal{H}_i^{\mathbf{k}}$ on sample $\boldsymbol{x}^k$ is then $(v_i - b_i(v_i))/c(k)$ if $c(k) > 0$ and 0 otherwise. The same utility is output on two samples $\boldsymbol{x}^k$ and $\boldsymbol{x}^{k'}$ whenever $c(k) = c(k')$. Therefore, if we look at the labels assigned to a set $S$ of samples that are mapped to the nonzero integer by $c$, there can be at most $|S| + 1 \leq m + 1$ patterns of labels, because we compare the same utility with $|S|$ witnesses; the set of samples mapped to 0 by $c$ have only one pattern of labels. The vector of labels generated by a hypothesis in such a sub-class is a concatenation of these patterns. The image of $c$ has $n$ nonzero integers, and so there are at most $(m+1)^n$ label vectors.

To conclude, the total number of label vectors generated by $\mathcal{H}_i = \bigcup_{\mathbf{k}} \mathcal{H}_i^{\mathbf{k}}$ is at most

$$(m+1)^{2(n-1)}(m+1)^n \leq (m+1)^{3n}.$$

To pseudo-shatter $m$ samples, we need $(m+1)^{3n} \geq 2^m$, which implies $m = O(n \log n)$.

$\square$

### B.1.2 Proof of Lemma 3.10

**Lemma 3.10.** *Let $\mathcal{H}$ be a class of functions from a product space $\boldsymbol{T}$ to $[0, H]$. If $\mathcal{H}$ is $(\epsilon, \delta)$-uniformly convergent with sample complexity $m = m(\epsilon, \delta)$, then $\mathcal{H}$ is $\left(2\epsilon, \frac{H\delta}{\epsilon}\right)$-uniformly convergent on product distribution with sample complexity $m$.*

*Proof.* Think of the samples $\boldsymbol{s}$ as an $m \times n$ matrix $(s_i^j)$, where each row $j$ represents sample $\boldsymbol{s}^j$, and each column $i$ consists of the values sampled from $F_i$. Then we draw $n$ permutations $\pi_1, \ldots, \pi_n$ of $[m] = \{1, \ldots, m\}$ independently and uniformly at random, and permute the $m$ elements in column $i$ by $\pi_i$. Regard each new row $j$ as a new sample, denoted by $\tilde{\boldsymbol{s}}^j = (s_1^{\pi_1(j)}, s_2^{\pi_2(j)}, \ldots, s_n^{\pi_n(j)})$. Given $\pi_1, \ldots, \pi_n$, the "permuted samples" $\tilde{\boldsymbol{s}}^j$, $j = 1, \ldots, m$ then have the same distributions as $m$ i.i.d. random draws from $\boldsymbol{F}$.

For $h \in \mathcal{H}$, let $p_h$ be $\mathbf{E}_{\boldsymbol{v} \sim \boldsymbol{F}}[h(\boldsymbol{v})]$. Then by the definition of $(\epsilon, \delta)$-uniform convergence (but not on product distribution),

$$\mathbf{Pr}_{\boldsymbol{s}, \pi}\left[\exists h \in \mathcal{H}, \left|p_h - \frac{1}{m}\sum_{j=1}^{m} h(\tilde{\boldsymbol{s}}^j)\right| \geq \epsilon\right] \leq \delta. \tag{12}$$

For a set of fixed samples $\boldsymbol{s} = (\boldsymbol{s}^1, \ldots, \boldsymbol{s}^m)$, recall that $E_i$ is the uniform distribution over $\{s_i^1, \ldots, s_i^m\}$, and $\boldsymbol{E} = \prod_{i=1}^{n} E_i$. We show that the expected value of $h$ on $\boldsymbol{E}$ satisfies $\mathbf{E}_{\boldsymbol{v} \sim \boldsymbol{E}}[h(\boldsymbol{v})] = \mathbf{E}_{\pi}[\frac{1}{m}\sum_{j=1}^{m} h(\tilde{\boldsymbol{s}}^j)]$. This is because

$$\begin{aligned}
\mathbf{E}_{\pi}\left[\frac{1}{m}\sum_{i=1}^{m} h(\tilde{\boldsymbol{s}}^j)\right] &= \frac{1}{m}\sum_{j=1}^{m}\mathbf{E}_{\pi}\left[h(\tilde{\boldsymbol{s}}^j)\right] \\
&= \frac{1}{m}\sum_{j=1}^{m}\sum_{(k_1, \ldots, k_n) \in [m]^n} h(s_1^{k_1}, \ldots, s_n^{k_n}) \cdot \\
&\qquad\qquad\qquad\qquad \mathbf{Pr}_{\pi}\left[\pi_1(j) = k_1, \ldots, \pi_n(j) = k_n\right] \\
&= \frac{1}{m}\sum_{j=1}^{m}\sum_{(k_1, \ldots, k_n) \in [m]^n} h(s_1^{k_1}, \ldots, s_n^{k_n}) \cdot \frac{1}{m^n} \\
&= \frac{1}{m^n}\sum_{(k_1, \ldots, k_n) \in [m]^n} h(s_1^{k_1}, \ldots, s_n^{k_n}) \\
&= \mathbf{E}_{\boldsymbol{v} \sim \boldsymbol{E}}\left[h(\boldsymbol{v})\right].
\end{aligned}$$

Thus,

$$\begin{aligned}
|p_h - \mathbf{E}_{\boldsymbol{v} \sim \boldsymbol{E}}\left[h(\boldsymbol{v})\right]| &= \left|p_h - \mathbf{E}_{\pi}\left[\frac{1}{m}\sum_{j=1}^{m} h(\tilde{\boldsymbol{s}}^j)\right]\right| \\
&\leq \mathbf{E}_{\pi}\left[\left|p_h - \frac{1}{m}\sum_{j=1}^{m} h(\tilde{\boldsymbol{s}}^j)\right|\right] \\
&\leq \mathbf{Pr}_{\pi}\left[\left|p_h - \frac{1}{m}\sum_{j=1}^{m} h(\tilde{\boldsymbol{s}}^j)\right| \geq \epsilon\right] \cdot H \\
&\qquad + \left(1 - \mathbf{Pr}_{\pi}\left[\left|p_h - \frac{1}{m}\sum_{j=1}^{m} h(\tilde{\boldsymbol{s}}^j)\right| \geq \epsilon\right]\right) \cdot \epsilon \\
&\leq \mathbf{Pr}_{\pi}\left[\mathrm{Bad}(h, \pi, \boldsymbol{s})\right] \cdot H + \epsilon,
\end{aligned}$$

where in the last step we define event

$$\mathrm{Bad}(h, \pi, \boldsymbol{s}) = \mathbb{I}\left[\left|p_h - \frac{1}{m}\sum_{j=1}^{m} h(\tilde{\boldsymbol{s}}^j)\right| \geq \epsilon\right].$$

By simple calculation, whenever $|p_h - \mathbf{E}_{\boldsymbol{v} \sim \boldsymbol{E}}[h(\boldsymbol{v})]| \geq 2\epsilon$, we have $\mathbf{Pr}_{\pi}[\mathrm{Bad}(h, \pi, \boldsymbol{s})] \geq \epsilon/H$.

Finally, consider the random draw $\boldsymbol{s} \sim \boldsymbol{F}$,

$$\begin{aligned}
\mathbf{Pr}_{\boldsymbol{s}}\left[\exists h \in \mathcal{H}, |p_h - \mathbf{E}_{\boldsymbol{v} \sim \boldsymbol{E}}\left[h(\boldsymbol{v})\right]| \geq 2\epsilon\right] &\leq \mathbf{Pr}_{\boldsymbol{s}}\left[\exists h \in \mathcal{H}, \mathbf{Pr}_{\pi}\left[\mathrm{Bad}(h, \pi, \boldsymbol{s})\right] \geq \frac{\epsilon}{H}\right] \\
&\leq \mathbf{Pr}_{\boldsymbol{s}}\left[\mathbf{Pr}_{\pi}\left[\exists h \in \mathcal{H}, \mathrm{Bad}(h, \pi, \boldsymbol{s}) \text{ holds}\right] \geq \frac{\epsilon}{H}\right].
\end{aligned}$$

By Markov's inequality, this is in turn upper bounded by

$$\frac{H}{\epsilon} \, \mathbf{E}_{\boldsymbol{s}} \left[ \mathbf{Pr}_{\pi} \left[ \exists h \in \mathcal{H}, \ \mathrm{Bad}(h, \pi, \boldsymbol{s}) \text{ holds} \right] \right] = \frac{H}{\epsilon} \, \mathbf{Pr}_{\boldsymbol{s}, \pi} \left[ \exists h \in \mathcal{H}, \ \mathrm{Bad}(h, \pi, \boldsymbol{s}) \text{ holds} \right]$$

$$\leq \frac{H\delta}{\epsilon} \qquad\qquad \text{By (12)}$$

$\square$

## B.2  Lower Bound: Proof of Theorem 3.15

**Theorem 3.15.** *For any $\epsilon < \frac{1}{4000}, \delta < \frac{1}{20}$, there is a family of product distributions for which no algorithm $(\epsilon, \delta)$-learns, with $m$ samples, utilities over the set of all monotone bidding strategies, for any $m \leq \frac{1}{4 \times 10^8} \cdot \frac{n}{\epsilon^2}$.*

Fixing $\epsilon > 0$, fixing $c_1 = 2000$, we first define two value distributions. Let $F^+$ be a distribution supported on $\{0, 1\}$, and for $v \sim F^+$, $\mathbf{Pr}[v = 0] = 1 - \frac{1 + c_1 \epsilon}{n}$, and $\mathbf{Pr}[v = 1] = \frac{1 + c_1 \epsilon}{n}$. Similarly define $F^-$: for $v \sim F^-$, $\mathbf{Pr}[v = 0] = 1 - \frac{1 - c_1 \epsilon}{n}$, and $\mathbf{Pr}[v = 1] = \frac{1 - c_1 \epsilon}{n}$.

Let $\mathrm{KL}(F^+; F^-)$ denote the KL-divergence between the two distributions.

**Claim B.1.** $\mathrm{KL}(F^+; F^-) = O(\frac{\epsilon^2}{n})$.

*Proof.* By definition,

$$\mathrm{KL}(F^+; F^-) = \frac{1 + c_1 \epsilon}{n} \ln\left( \frac{1 + c_1 \epsilon}{1 - c_1 \epsilon} \right) + \frac{n - 1 - c_1 \epsilon}{n} \ln\left( \frac{n - 1 - c_1 \epsilon}{n - 1 + c_1 \epsilon} \right)$$

$$= \frac{1}{n} \ln\left( \frac{1 + c_1 \epsilon}{1 - c_1 \epsilon} \cdot \frac{(1 - \frac{c_1 \epsilon}{n-1})^{n-1}}{(1 + \frac{c_1 \epsilon}{n-1})^{n-1}} \right) + \frac{c_1 \epsilon}{n} \ln\left( \frac{1 + c_1 \epsilon}{1 - c_1 \epsilon} \cdot \frac{1 + \frac{c_1 \epsilon}{n-1}}{1 - \frac{c_1 \epsilon}{n-1}} \right)$$

$$\leq \frac{1}{n} \ln\left( \frac{1 + c_1 \epsilon}{1 - c_1 \epsilon} \cdot \frac{\left(1 - \frac{c_1 \epsilon}{n-1}\right)^{n-1}}{1 + c_1 \epsilon} \right) + \frac{2c_1 \epsilon}{n} \ln\left( 1 + \frac{2c_1 \epsilon}{1 - c_1 \epsilon} \right)$$

$$\leq \frac{1}{n} \ln\left( \frac{1 - c_1 \epsilon + \frac{1}{2}(c_1 \epsilon)^2}{1 - c_1 \epsilon} \right) + \frac{8c_1^2 \epsilon^2}{n}$$

$$\leq \frac{10 c_1^2 \epsilon^2}{n}.$$

In the last two inequalities we used $c_1 \epsilon < \frac{1}{2}$ and $\ln(1 + x) \leq 1 + x$ for all $x > 0$. $\qquad\square$

It is well known that upper bounds on KL-divergence implies information theoretic lower bound on the number of samples to distinguish distributions (e.g. Mansour, 2011).

**Corollary B.2.** *Given $t$ i.i.d. samples from $F^+$ or $F^-$, if $t \leq \frac{n}{80 c_1^2 \epsilon^2}$, no algorithm $\mathcal{H}$ that maps samples to $\{F^+, F^-\}$ can do the following: when the samples are from $F^+$, $\mathcal{H}$ outputs $F^+$ with probability at least $\frac{2}{3}$, and if the samples are from $F^-$, $\mathcal{H}$ outputs $F^-$ with probability at least $\frac{2}{3}$.*

We now construct product distributions using $F^+$ and $F^-$. For any $S \subseteq [n - 1]$, define product distribution $\boldsymbol{F}_S$ to be $\prod_i F_i$ where $F_i = F^+$ if $i \in S$, and $F_i = F^-$ if $i \in [n - 1] \setminus S$, and $F_n$ is a point mass on value 1. For any $j \in [n - 1]$ and $S \subseteq [n - 1]$, distinguishing $\boldsymbol{F}_{S \cup \{j\}}$ and $\boldsymbol{F}_{S \setminus \{j\}}$ by samples from the product distribution is no easier than distinguishing $F^+$ and $F^-$, because the coordinates of the samples not from $F_j$ contains no information about $F_j$.

**Corollary B.3.** *For any $j \in [n - 1]$ and $S \subseteq [n - 1]$, given $t$ i.i.d. samples from $\boldsymbol{F}_{S \cup \{j\}}$ or $\boldsymbol{F}_{S \setminus \{j\}}$, if $t \leq \frac{n}{80 c_1^2 \epsilon^2}$, no algorithm $\mathcal{H}$ can do the following: when the samples are from $\boldsymbol{F}_{S \cup \{j\}}$, $\mathcal{H}$ outputs $\boldsymbol{F}_{S \cup \{j\}}$ with probability at least $\frac{2}{3}$, and when the samples are from $\boldsymbol{F}_{S \setminus \{j\}}$, $\mathcal{H}$ outputs $\boldsymbol{F}_{S \setminus \{j\}}$ with probability at least $\frac{2}{3}$.*

We now use Corollary B.3 to derive an information theoretic lower bound on learning utilities for monotone bidding strategies, for distributions in $\{\boldsymbol{F}_S\}_{S \subseteq [n]}$.

*Proof of Theorem 3.15.* Without loss of generality, assume $n$ is odd. Let $S$ be an arbitrary subset of $[n-1]$ of size either $\lfloor n/2 \rfloor$ or $\lceil n/2 \rceil$. We focus on the interim utility of bidder $n$ with value $1$ and bidding $\frac{1}{2}$. Denote this bidding strategy by $b_n(\cdot)$. The other bidders may adopt one of two bidding strategies. One of them is $b^+(\cdot)$: $b^+(0) = 0$ and $b^+(1) = \frac{1}{2} + \eta$ for sufficiently small $\eta > 0$. The other bidding strategy $b^-(\cdot)$ maps all values to $0$. For $T \subseteq [n-1]$, let $\boldsymbol{b}_T(\cdot)$ be the profile of bidding strategies where $b_i(\cdot) = b^+(\cdot)$ for $i \in T$, and $b_i(\cdot) = b^-(\cdot)$ for $i \notin T$.

For the distribution $\boldsymbol{F}_S$,

$$
u_n\left(1, \frac{1}{2}, \boldsymbol{b}_T(\cdot)\right) = \frac{1}{2} \mathbf{Pr}\left[\max_{i \in T} v_i = 0\right]
$$

$$
= \frac{1}{2}\left(1 - \frac{1 + c_1\epsilon}{n}\right)^{|S \cap T|}\left(1 - \frac{1 - c_1\epsilon}{n}\right)^{|T \setminus S|}
$$

$$
= \frac{1}{2}\left(1 - \frac{1 + c_1\epsilon}{n}\right)^{|T|}\left(\frac{n - 1 + c_1\epsilon}{n - 1 - c_1\epsilon}\right)^{|T \setminus S|}.
$$

Therefore, for $T, T' \subseteq [n-1]$ with $|T| = |T'|$,

$$
\frac{u_n(1, \frac{1}{2}, \boldsymbol{b}_T(\cdot))}{u_n(1, \frac{1}{2}, \boldsymbol{b}_{T'}(\cdot))} = \left(1 + \frac{2c_1\epsilon/(n-1)}{1 - \frac{c_1\epsilon}{n-1}}\right)^{|T \setminus S| - |T' \setminus S|}
$$

$$
\geq 1 + \frac{2c_1\epsilon}{n-1} \cdot (|T \setminus S| - |T' \setminus S|);
$$

Suppose $|T \setminus S| \geq |T' \setminus S|$ and $|T| = |T'| \geq \lfloor \frac{n}{2} \rfloor$, then

$$
u_n\left(1, \frac{1}{2}, \boldsymbol{b}_T(\cdot)\right) - u_n\left(1, \frac{1}{2}, \boldsymbol{b}_{T'}(\cdot)\right) \geq (|T \setminus S| - |T' \setminus S|) \cdot \frac{2c_1\epsilon}{n-1} \cdot u_n\left(1, \frac{1}{2}, \boldsymbol{b}_{T'}(\cdot)\right)
$$

$$
\geq (|T \setminus S| - |T' \setminus S|) \cdot \frac{2c_1\epsilon}{n-1} \cdot \frac{1}{8e^2}, \tag{13}
$$

where the last inequality is because $u_n(1, \frac{1}{2}, \boldsymbol{b}_{T'}(\cdot)) \geq \frac{1}{2}(1 - \frac{2}{n})^n = \frac{1}{2}[(1 - \frac{2}{n})^{\frac{n}{2}}]^2 \geq \frac{1}{2}(\frac{1}{2e})^2 = \frac{1}{8e^2}$.

Now suppose an algorithm $\mathcal{A}$ $(\epsilon, \delta)$-learns the utilities of all monotone bidding strategies with $t$ samples $\boldsymbol{s}$ for $t \leq \frac{n}{80c_1^2\epsilon^2}$. Define $\mathcal{H} : \mathbb{R}_+^{n \times t} \times \mathbb{N} \to 2^{[n-1]}$ be a function that outputs among all $T \subseteq [n-1]$ of size $k$, the one that maximizes bidder $n$'s utility when they bid according to bidding strategy $\boldsymbol{b}_T$. Formally,

$$
\mathcal{H}(\boldsymbol{s}, k) = \underset{T \subseteq [n-1], |T| = k}{\arg\max} \mathcal{A}\left(\boldsymbol{s}, n, 1, (\boldsymbol{b}_T(\cdot), b_n(\cdot))\right),
$$

By Definition 3.1, for any $S$ with $|S| = \lfloor n/2 \rfloor$, for samples drawn from $\boldsymbol{F}_S$, with probability at least $1 - \delta$,

$$
\mathcal{A}(\boldsymbol{s}, n, 1, (\boldsymbol{b}_{[n-1] \setminus S}(\cdot), b_n(\cdot)) \geq u_n\left(1, \frac{1}{2}, \boldsymbol{b}_{[n-1] \setminus S}(\cdot)\right) - \epsilon;
$$

and for any $T \subseteq [n-1]$ with $|T| = \lceil n/2 \rceil$,

$$
\mathcal{A}(\boldsymbol{s}, n, 1, (\boldsymbol{b}_T(\cdot), b_n(\cdot)) \leq u_n\left(1, \frac{1}{2}, \boldsymbol{b}_T(\cdot)\right) + \epsilon.
$$

Therefore, for $W = \mathcal{H}(\boldsymbol{s}, \lceil n/2 \rceil)$,

$$
u_n\left(1, \frac{1}{2}, \boldsymbol{b}_W(\cdot)\right) \geq u_n\left(1, \frac{1}{2}, \boldsymbol{b}_{[n-1] \setminus S}(\cdot)\right) - 2\epsilon.
$$

Since $|W| = [n-1] \setminus S = \lceil n/2 \rceil$, by (13),

$$
\left(\lceil \frac{n}{2} \rceil - |W \setminus S|\right) \cdot \frac{c_1\epsilon}{(n-1)4e^2} \leq 2\epsilon.
$$

594 So

$$|W \cap S| \leq (n-1) \cdot \frac{8e^2}{c_1}.$$

595 In other words, with probability at least $1 - \delta$, $\mathcal{H}(\boldsymbol{s}, \lceil n/2 \rceil)$ is the complement of $S$ except for at
596 most $\frac{8e^2}{c_1}$ fraction of the coordinates in $[n-1]$.

597 Similarly, for $S$ of cardinality $\lceil n/2 \rceil$,

$$|\mathcal{H}(\boldsymbol{s}, \lceil n/2 \rceil) \cap S| \leq (n-1) \cdot \frac{8e^2}{c_1} + 1.$$

598 Take $c_2$ to be $\frac{8e^2}{c_1}$. We have $c_2 < \frac{1}{20}$. For all large enough $n$ and all $S$ of size $\lfloor n/2 \rfloor$ or $\lceil n/2 \rceil$, with
599 probability at least $1 - \delta$, $\mathcal{H}(\boldsymbol{s}, \lceil n/2 \rceil)$ correctly outputs the elements not in $S$ with an exception of
600 at most $c_2$ fraction of coordinates.

601 Let $\mathcal{S}$ be the set of all subsets of $[n-1]$ of size either $\lceil n/2 \rceil$ or $\lfloor n/2 \rfloor$. Consider any $S \in \mathcal{S}$. Let
602 $\theta(S) \subseteq [n-1]$ denote the set of coordinates whose memberships in $S$ are correctly predicted
603 by $\mathcal{H}(\boldsymbol{s}, \lceil n/2 \rceil)$ with probability at least $2/3$; that is, $i \in \theta(S)$ iff with probability at least $2/3$,
604 $\mathcal{H}(\boldsymbol{s}, \lceil n/2 \rceil)$ is correct about whether $i \in S$. Let the cardinality of $|\theta(S)|$ be $z(n-1)$. Suppose we
605 draw coordinate $i$ uniformly at random from $[n-1]$, and independently draw samples $\boldsymbol{s}$ from $\boldsymbol{F}_S$,
606 then the probability that $\mathcal{H}(\boldsymbol{s}, \lceil n/2 \rceil)$ is correct about whether $i \in S$ satisfies:

$$\mathbf{Pr}_{i,\boldsymbol{s}} \left[ \mathcal{H}(\boldsymbol{s}, \lceil n/2 \rceil) \text{ is correct about whether } i \in S \right] \geq (1 - c_2)(1 - \delta)$$
$$\geq 0.9,$$

607 and

$$\mathbf{Pr}_{i,\boldsymbol{s}} \left[ \mathcal{H}(\boldsymbol{s}, \lceil n/2 \rceil) \text{ is correct about whether } i \in S \right] \leq \mathbf{Pr}_i \left[ i \in \theta(S) \right] \cdot 1 + \mathbf{Pr}_i \left[ i \notin \theta(S) \right] \cdot \frac{2}{3}$$
$$= z \cdot 1 + (1 - z) \cdot \frac{2}{3},$$

608 which implies $z > 0.6$. If a pair of sets $S$ and $S'$ differ in only one coordinate $i$, and $i \in \theta(S) \cap \theta(S')$,
609 then $\mathcal{H}(\cdot)$ serves as an algorithm that tells apart $\boldsymbol{F}_S$ and $\boldsymbol{F}_{S'}$, contradicting Corollary B.3. We now
610 show, with a counting argument, that such a pair of $S$ and $S'$ must exist.

611 Since for each $S \in \mathcal{S}$, $|\theta(S)| \geq 0.6(n-1)$, there exists a coordinate $i \in [n-1]$ and $\mathcal{T} \subseteq \mathcal{S}$, with
612 $|\mathcal{T}| \geq 0.6|\mathcal{S}|$, such that for each $S \in \mathcal{T}$, $i \in \theta(S)$. But $\mathcal{S}$ can be decomposed into $|\mathcal{S}|/2$ pairs of
613 sets, such that within each pair, the two sets differ by one in size, and precisely one of them contains
614 coordinate $i$. Therefore among these pairs there must exist one $(S, S')$ with $S, S' \in \mathcal{T}$, i.e., $i \in \theta(S)$
615 and $i \in \theta(S')$. Using $\mathcal{H}$, which is induced by $\mathcal{A}$, we can tell apart $\boldsymbol{F}_S$ and $\boldsymbol{F}_{S'}$ with probability at
616 least $2/3$, which is a contradiction to Corollary B.3. This completes the proof of Theorem 3.15. $\square$

## C Auctions with Costly Search

618 We extend our sample complexity results to auctions in which bidders need to incur a cost to know
619 precisely their values, a model proposed and studied by Kleinberg et al. (2016).

620 In this model, each bidder $i$ knows the distribution $F_i$ from which her value is drawn, but gets to
621 know her value $v_i$ only after incurring a cost $c_i$. This models well, for example, a real estate market,
622 where $c_i$ is an inspection cost. Kleinberg et al. (2016) showed that, due to the search costs, the
623 English auction can have low efficiency, whereas the Dutch auction, with its descending price, can
624 coordinate the bidders' searching in an almost efficient way. Intuitively, a bidder does not inspect
625 her value until the price drops to a certain level, and then, after inspection at this threshold, either
626 claims the item at the threshold price, or waits till later. In fact, absent incentive issues, this is the
627 procedure a central authority would follow to maximize the welfare; the elegant algorithm is known
628 as the Pandora's Box algorithm (Weitzman, 1979). With incentives, bidders shade their bids just
629 as in a first price auction, and there is efficiency loss. This was made precise by Kleinberg et al.,
630 who showed a correspondence between the equilibria in a Dutch auction with search costs and the
631 equilibria in a first price auction without search costs but with transformed value distributions. The

near efficiency of the Dutch auction therefore follows from Price of Anarchy results on the first price auction (Syrgkanis and Tardos, 2013; Hoy et al., 2018).

In this appendix, we first review in Section C.1 Pandora's Box algorithm, necessary for understanding the correspondence observed by Kleinberg et al. (2016). En route, we show that $\tilde{O}(n/\epsilon^2)$ samples from the value distributions suffice for the algorithm to be $\epsilon$-close to optimal when the distributions are unknown. Our bound slightly improves a recent result by Guo et al. (2019a).

We then review, in Section C.2, the correspondence between the Dutch auction with search costs and the FPA without search costs. The correspondence between auctions involves mappings between strategies and a transformation on value distributions. These mappings and transformation depend on the value distributions. We show that, when the value distributions are unknown, with $\tilde{O}(1/\epsilon^2)$ value samples, an "empirical correspondence" can be established such that all monotone bidding strategies in the Dutch auction have approximately the same utilities as the corresponding bidding strategies in an FPA; combining with our learning results on the FPA, with $\tilde{O}(n/\epsilon^2)$ samples, any equilibrium of the FPA without search costs on a transformed empirical distribution can be mapped to an approximate equilibrium of the Dutch auction on the true distribution.

## C.1 Pandora's Box Problem and Its Sample Complexity

Absent search costs, the welfare (a.k.a. the efficiency) of a single item auction is the value of the bidder who is allocated the item. The maximum expected welfare is therefore simply the expectation of the largest value among the bidders. Auctions that sell to the highest bidder and charges the winner a price equal to the second highest bid gives bidders correct incentives to bid their true values and maximizes the welfare. The sealed-bid second price auction, the ascending price auction (English auction) and the descending price auction (Dutch auction) all achieve this. With search costs, the welfare of an auction is the value of the bidder winning the item minus all the search costs paid. Even without incentive considerations, the problem is nontrivial algorithmically.

**The Pandora's Box Problem.** The following Pandora's Box problem, named by Weitzman (1979), abstracts the welfare maximization problem in the presence of search costs. We are given $n$ boxes, each box $i$ containing a value $v_i$ drawn independently from a known distribution $F_i$; to open box $i$ and see $v_i$, we must pay a cost of $c_i$; at any point, we can take any box that has been opened and quit, or open a closed box at a cost, or quit without taking anything. Our payoff is the value in the box taken (if any) minus the costs we paid along the way. Given $F_1, \cdots, F_n$ and $c_1, \ldots, c_n$, we need to compute a procedure that maximizes the expected payoff.

Weitzman (1979) used this setting to model a consumer searching for an item to purchase; he gave an optimal algorithm, which is in turn a special case of Gittins Index algorithm from Bayesian bandits (Gittins, 1979).

We describe his algorithm below. To facilitate discussion of learning, we treat search costs as given, and algorithms as mappings from (unseen) values $v_1, \ldots, v_n$ to a payoff. Certainly, only mappings that correspond to valid search procedures are meaningful; in particular, the procedure's decision (e.g., to open which box) cannot depend on values that have not been revealed. It is the associated search procedure that we are interested in.

**Definition C.1** (Index Based Algorithms/Mappings). *Given search costs* $(c_1, \ldots, c_n)$*, a mapping* $\mathcal{A}$ *from* $(v_1, \ldots, v_n) \in [0, H]^n$ *to* $\mathbb{R}$ *is* index based *if there exist* indices $r_1, \ldots, r_n \in \mathbb{R}$ *such that on any vector of values* $(v_1, \ldots, v_n)$*, the output of* $\mathcal{A}$ *is given by the following procedure:*

1. *Initialize: let the current option be* 0 *(for taking nothing), write* $r_i$ *on box* $i$ *for* $i = 1, \ldots, n$, *and let the cumulative cost be* 0.

2. *Iterate till termination:*
   *If all the numbers written on the box are lower than the current option:*

   - *Stop searching, and output the current option minus the cumulative cost.*

   *Otherwise:*

   - *Let box* $i$ *be the box with the largest number written on it.*
   - *If the number written on box* $i$ *is a value* $(v_i)$*, then replace the current option by* $v_i$.

- *If the number written on box $i$ is an index ($r_i$), then open box $i$, add $c_i$ to the cumulative cost, reveal $v_i$ and replace the number written on box $i$ by $v_i$.*

**Theorem C.2** (Weitzman, 1979). *The optimal algorithm corresponds to an index-based mapping; the index $r_i$ for box $i$ is the unique solution to $\mathbf{E}_{v \sim F_i}[\max(v - r_i, 0)] = c_i$.*

**Learning.** We now answer the following learning question: if the distributions $F_1, \cdots, F_n$ are unknown, how many samples from them suffice for us to devise an algorithm that is close to optimal on the original distribution? Recently, Guo et al. (2019a) gave a polynomial bound for the problem; we give an alternative analysis using pseudo-dimension, which leads to a slightly improved bound. We make use of a technical lemmas of theirs (Lemma C.5). For our learning algorithm to be run in polynomial time, we invoke Lemma 3.10 to perform learning on the empirical product distribution.

Given our view of the algorithms as mappings from value vectors to the payoff, the expected payoff of an algorithm is then the expectation of its output on the value distributions. Given Theorem C.2, it suffices to learn the expected payoff of all index-based algorithms. The problem then boils down to bounding the pseudo-dimension of the class of index-based mappings. Modulo a technical issue which calls for truncating the index-based algorithms, that is an outline of the proof of the following sample complexity theorem.

**Theorem C.3.** *Given search costs $c_1, \ldots, c_n$, such that for any $\epsilon, \delta \in (0, 1)$, there is $M = O\left(\frac{H^2 n \log n}{\epsilon^2} \log^2(\frac{1}{\epsilon}) \left[\log(\frac{H}{\epsilon}) + \log(\frac{H}{\epsilon \delta})\right]\right)$, such that for any $m > M$, given $m$ samples, a search procedure computed on these samples has expected payoff within additive $\epsilon$ to the optimal algorithm with probability at least $1 - \delta$. Moreover, the procedure can be computed in polynomial time.*

We devote the rest of this subsection to the proof of this theorem. Let $\mathcal{H}_P$ be the class of all index-based mappings. The technical centerpiece is a bound on the pseudo-dimension of $\mathcal{H}_P$.

**Lemma C.4.** $\mathrm{Pdim}(\mathcal{H}_P) = O(n \log n)$.

*Proof.* Given any profile of values $(v_1, \ldots, v_n) \in [0, H]^n$, the output of any index-based mapping with indices $(r_i)_i$ is fully determined by the following $O(n^2)$ linear inequalities: for any $i, j \in [n]$, whether $r_i \geq r_j$ or $r_i < r_j$; for any $i, j \in [n]$, whether $r_i \geq v_j$ or $r_i < v_j$. That is, the space of indices is partitioned by the hyperplanes given by these $O(n^2)$ inequalities, and within each region the corresponding index-based mapping remains a constant for this profile of values. Consider any $m$ value profiles that are pseudo-shattered by $\mathcal{H}_P$. Each of these $m$ value profiles imposes $O(n^2)$ linear inequalities on the space of indices, and we will have altogether $O(mn^2)$ inequalities. A crucial observation is that, for any positive integer $t$, the space $\mathbb{R}^n$ can be partitioned by $t$ hyperplanes into at most $O(t^n)$ regions. Therefore the space of indices, which is $\mathbb{R}^n$, can be divided into at most $(Cmn^2)^n$ regions, for some constant $C > 0$. Any index-based algorithm within such a region gives the same outputs on all these $m$ value profiles, and therefore cannot give different signs for any profile no matter what the corresponding witness is. To shatter $m$ profiles we need at least $2^m$ regions. Therefore $2^m \leq (Cmn^2)^n$, which gives $m \leq C'n \log n$ for some $C' > 0$. $\qquad\square$

Note that, if the values are between 0 and $H$, without loss of generality we may assume $c_i \leq H$ for each $i$. (Otherwise the box should be discarded by any reasonable algorithm.) With this, directly combining Lemma C.4 and Theorem 3.6 would still yield a bound having a cubic dependence on $n$, because the output of an index-based mapping may span the range $[-nH, H]$. A similar problem also arose in the approach of Guo et al. (2019a), who remedied this by observing that the performance of the optimal index algorithm is not affected much if it is truncated: to *truncate* an algorithm for the Pandora's Box problem, the algorithm is terminated immediately when its cumulated cost exceeds $\Omega(\log \frac{1}{\epsilon})$.

**Lemma C.5** (Lemma 25 of Guo et al., 2019a). *On an instance of the Pandora's Box problem, the expected payoff of the optimal index-based algorithm exceeds that of its truncated version by no more than $\epsilon$.*

The proof of Lemma C.4 is easily modified to give the same bound on the pseudo-dimension of mappings corresponding to truncated index-based algorithms. With this, we can now combine Theorem 3.6 and Lemma 3.10 to obtain a sample complexity upper bound.

Compared with Guo et al. (2019a)'s bound $O(\frac{n}{\epsilon^2} \log^2(\frac{1}{\epsilon}) \log(\frac{n}{\epsilon}) \log(\frac{n}{\epsilon\delta}))$ (where $H$ is normalized to 1), our bound is better: theirs has a $\frac{n}{\epsilon^2} \log^2(\frac{1}{\epsilon})(\log^2 n + \log\frac{1}{\epsilon} \log\frac{1}{\epsilon\delta})$ term while we do not.

We remark that in Theorem C.3 we show the sample complexity for uniform convergence *on product distribution*, because this yields a fast algorithm given samples: simply running the optimal truncated index-based algorithm on the empirical product distribution is guaranteed to be approximately optimal on $\boldsymbol{F}$ with high probability. On the other hand, picking out the best index-based algorithm on the empirical distribution, which is correlated, appears computationally challenging.

## C.2 Descending Auction with Search Costs

In this section, we briefly review the main results by Kleinberg et al. (2016) in Section C.2.1, and then in Section C.2.2 present our learning results in auctions with search costs. Recall that in this setting, we consider a single-item auction, where each bidder $i$ has a value $v_i \in [0, H]$ drawn independently from distribution $F_i$, but $v_i$ is not known to anyone at the beginning of the auction. In order to observe the value, bidder $i$ needs to pay a known search cost $c_i \in [0, H]$.

### C.2.1 Transformation with Distributional Knowledge

**Descending auction with search costs.** In a *descending auction* (or Dutch auction), a publicly visible price descends continuously from $H$. At any point, any bidder may claim the item at the current price. With search cost, a bidder's strategy $\alpha_i$ consists of two parts:[2] a threshold price $t_i$ and a mapping $b_i(\cdot)$ from values to bids. Concretely, bidder $i$ decides to inspect when the price descends to $t_i$, at which point she pays the search cost and immediately learns her value $v_i$. After seeing her value, the bidder chooses another a purchase price $b_i(v_i) \leq t_i$ at which to claim the item. The latter is equivalent to submitting a bid $b_i(v_i) \leq t_i$.

We say a strategy $\alpha_i = (t_i, b_i(\cdot))$ is *monotone* if $b_i(\cdot)$ is monotone non-decreasing. A strategy is *mixed* if it is a distribution over pure strategies $\alpha_i$'s. Mixed strategies allow bidders to randomize over the threshold price $t_i$ and the purchase price $b_i(v_i)$. Abusing notations, we also use $\alpha_i$ to denote a mixed strategy. We say a *mixed* strategy $\alpha_i$ is *monotone* if it is a distribution over monotone pure strategies.

We use $\mathrm{DA}(\boldsymbol{F}, \boldsymbol{c})$ to denote a descending auction on value distributions $\boldsymbol{F}$ with search costs $\boldsymbol{c}$, and let $u_i^{\mathrm{DA}(\boldsymbol{F},\boldsymbol{c})}(\alpha_i, \boldsymbol{\alpha}_{-i})$ be the expected utility of bidder $i$ when bidders use strategies $\boldsymbol{\alpha} = (\alpha_i, \boldsymbol{\alpha}_{-i})$ and their values are drawn from $\boldsymbol{F}$. Note that this utility is ex ante, since the value is unknown until the bidder searches. The solution concept we consider is therefore a Nash equilibrium rather than a Bayes Nash equilibrium.

**Definition C.6.** *In* $\mathrm{DA}(\boldsymbol{F}, \boldsymbol{c})$*, a (mixed) strategy profile $\boldsymbol{\alpha}$ is an $\epsilon$-Nash equilibrium (NE) if for each bidder $i$ and any strategy $\alpha_i'$,*

$$u_i^{\mathrm{DA}(\boldsymbol{F},\boldsymbol{c})}(\alpha_i', \boldsymbol{\alpha}_{-i}) - u_i^{\mathrm{DA}(\boldsymbol{F},\boldsymbol{c})}(\alpha_i, \boldsymbol{\alpha}_{-i}) \leq \epsilon.$$

*If $\epsilon = 0$, $\boldsymbol{\alpha}$ is a Nash equilibrium.*

We use $\mathrm{FPA}(\boldsymbol{F})$ to denote the first price auction with value distributions $\boldsymbol{F}$. Denote by $u_i^{\mathrm{FPA}(\boldsymbol{F})}(\boldsymbol{\beta})$ the (ex ante) expected utility of bidder $i$ in $\mathrm{FPA}(\boldsymbol{F})$, when the bidders use strategy profile $\boldsymbol{\beta}$. We can similarly define the Nash equilibrium for a first price auction.

**Definition C.7.** *In* $\mathrm{FPA}(\boldsymbol{F})$*, a (mixed) strategy profile $\boldsymbol{\beta}$ is an $\epsilon$-Nash equilibrium (NE) if for each bidder $i$ and any strategy $\beta_i'$,*

$$u_i^{\mathrm{FPA}(\boldsymbol{F})}(\beta_i', \boldsymbol{\beta}_{-i}) - u_i^{\mathrm{FPA}(\boldsymbol{F})}(\beta_i, \boldsymbol{\beta}_{-i}) \leq \epsilon.$$

*If $\epsilon = 0$, $\boldsymbol{\beta}$ is a Nash equilibrium.*

Note that Nash equilibrium is an ex ante notion, in contrast with BNE (Definition 2.1), which is an interim notion and requires that every type best respond. In $\mathrm{FPA}(\boldsymbol{F})$, an $\epsilon$-BNE must be an $\epsilon$-NE, but the reverse is not true.

775 With no search cost, the descending auction is well known to be equivalent to a first price auction.
776 Kleinberg et al. (2016) gave a first price auction without search costs and with transformed value
777 distributions, and showed that the NE of this auction corresponds to the NE of the Dutch auction with
778 search costs.

**Definition C.8.** *Given a distribution $F_i$ and a search cost $c_i$, define the index $r_i$ of $(F_i, c_i)$ to be*
780 *the unique solution to $\mathbf{E}_{v_i \sim F_i}[\max\{v_i - r_i, 0\}] = c_i$. If $c_i = 0$, let $r_i = H$. Always assume*
781 *$\mathbf{E}_{v_i \sim F_i}[v_i] \geq c_i$, so that $r_i \in [0, H]$. (Otherwise the search cost would be so high that the bidder*
782 *should never search for the value.)*

783 For a distribution $F$ and $r \in \mathbb{R}$, denote by $F^r$ the distribution of $\kappa := \min\{v, r\}$ where $v \sim F$. For a
784 product distribution $\boldsymbol{F}$ and a vector $\boldsymbol{r}$, we use $\boldsymbol{F^r}$ to denote the product distribution where the $i$-th
785 component is $F_i^{r_i}$. A key insight of Kleinberg et al. (2016) is a pair of utility-preserving mappings
786 between strategies in DA$(\boldsymbol{F}, \boldsymbol{c})$ and FPA$(\boldsymbol{F^r})$, where $\boldsymbol{r}$ is the vector of indices for $(\boldsymbol{F}, \boldsymbol{c})$.

**Definition C.9.** *For each bidder $i$, given distribution $F_i$ and $r_i \in [0, H]$, define two mappings:*[3]

1. *$\lambda^{r_i}$: for a monotone strategy $\beta_i : [0, r_i] \to \mathbb{R}_+$ for FPA$(\boldsymbol{F^r})$, its image strategy $\lambda^{\boldsymbol{r}}(\beta_i)$*
789 *in DA$(\boldsymbol{F}, \boldsymbol{c})$ consists of the threshold price $t_i = \beta_i(r_i)$ and the bidding function $b_i(v_i) = $*
790 *$\beta_i(\min\{v_i, r_i\})$. (By the monotonicity of $\beta_i$, we have $b_i(v_i) \leq t_i$).*

2. *$\mu^{(F_i, r_i)}$: for a strategy $\alpha_i = (t_i, b_i(\cdot))$ for DA$(\boldsymbol{F}, \boldsymbol{c})$, its image strategy $\beta_i = \mu^{(F_i, r_i)}(\alpha_i)$*
792 *in FPA$(\boldsymbol{F^r})$ is defined as $\beta_i(v_i) = b_i(v_i)$ for $\kappa_i < r_i$ and $\beta_i(r_i) = b_i(v_i')$ for a $v_i'$ redrawn*
793 *from $F_i$, conditioning on $v_i' \geq r_i$.*

794 The superscripts $r_i$ and $(F_i, r_i)$ should make it clear that the mapping $\lambda^{r_i}$ is determined solely by $r_i$
795 while $\mu^{(F_i, r_i)}$ is related to both the distribution and $r_i$.

796 We say a strategy $\alpha_i$ in a descending auction *claims above $r_i$* if $v_i \geq r_i \implies b_i(v_i) = t_i$, i.e., the
797 bidder claims the item immediately if she finds the value of the item greater than or equal to $r_i$.

**Claim C.10** (Claim 2 of Kleinberg et al., 2016). *Given distribution $F_i$ and index $r_i$,*

1. *If $\alpha_i$ claims above $r_i$, then $\alpha_i = \lambda^{r_i}(\mu^{(F_i, r_i)}(\alpha_i))$.*

2. *If $\beta_i$ is monotone, then $\beta_i = \mu^{(F_i, r_i)}(\lambda^{r_i}(\beta_i))$.*

**Theorem C.11** (Claim 3 of Kleinberg et al., 2016). *Suppose $\boldsymbol{r}$ is the indices of $(\boldsymbol{F}, \boldsymbol{c})$ (Definition C.8).*

1. *For any monotone mixed strategy profile $\boldsymbol{\beta} = (\beta_i, \boldsymbol{\beta}_{-i})$ for FPA$(\boldsymbol{F^r})$, for each bidder $i$,*

$$u_i^{\text{FPA}(\boldsymbol{F^r})}(\boldsymbol{\beta}) = u_i^{\text{DA}(\boldsymbol{F}, \boldsymbol{c})}(\lambda^{\boldsymbol{r}}(\boldsymbol{\beta})).$$

2. *For any mixed (not necessarily monotone) strategy profile $\boldsymbol{\alpha} = (\alpha_i, \boldsymbol{\alpha}_{-i})$ for DA$(\boldsymbol{F}, \boldsymbol{c})$, for*
804 *each bidder $i$,*
$$u_i^{\text{DA}(\boldsymbol{F}, \boldsymbol{c})}(\boldsymbol{\alpha}) \leq u_i^{\text{FPA}(\boldsymbol{F^r})}(\mu^{(\boldsymbol{F}, \boldsymbol{r})}(\boldsymbol{\alpha})),$$
805 *where "=" holds if $\alpha_i$ claims above $r_i$.*

**Theorem C.12** (Kleinberg et al., 2016). *Given DA$(\boldsymbol{F}, \boldsymbol{c})$ and FPA$(\boldsymbol{F^r})$ where $\boldsymbol{r}$ is the indices of*
807 *$(\boldsymbol{F}, \boldsymbol{c})$. If $\boldsymbol{\beta}$ is a BNE in FPA$(\boldsymbol{F^r})$, then $\lambda^{\boldsymbol{r}}(\boldsymbol{\beta})$ is an NE in DA$(\boldsymbol{F}, \boldsymbol{c})$. Conversely, if $\boldsymbol{\alpha}$ is an NE in*
808 *DA$(\boldsymbol{F}, \boldsymbol{c})$, then $\mu^{(\boldsymbol{F}, \boldsymbol{r})}(\boldsymbol{\alpha})$ is an NE in FPA$(\boldsymbol{F^r})$.*

### C.2.2 Transformation with Samples

810 We are now ready to present our learning results in auctions with search costs. In Kleinberg et al.
811 (2016), the utility- and equilibrium-preserving mappings $\lambda^{\boldsymbol{r}}$ and $\mu^{(\boldsymbol{F}, \boldsymbol{r})}$ are distribution-dependent.
812 We examine the number of samples needed to compute approximations of these mappings, when
813 the value distributions are unknown. We find that, given search costs and value samples, $\tilde{O}(1/\epsilon^2)$
814 samples suffice to construct mappings between strategies that approximately preserve utility; with
815 $\tilde{O}(n/\epsilon^2)$ samples, any equilibrium of the first price auction without search costs on a transformed

816 empirical distribution can be mapped to an approximate equilibrium of the descending auction on the
817 true distribution.

818 When value distribution $F_i$'s are unknown (but cost $c_i$'s are known), the mapping $\lambda^{\boldsymbol{r}}$ cannot be used
819 to transform an NE for a first price auction with no search costs to an $\epsilon$-NE for a descending auction
820 with search costs because the computation of index $r_i$ involves distribution $F_i$. Instead, we estimate
821 an index $\hat{r}_i$ from samples and use the corresponding mapping $\lambda^{\hat{\boldsymbol{r}}}$ to do so.

822 **Definition C.13.** *Partition the samples $\boldsymbol{s}$ into two sets, $\boldsymbol{s}^A$ and $\boldsymbol{s}^B$, each of size $m/2$. Denote the*
823 *empirical product distributions on $\boldsymbol{s}^A$ and $\boldsymbol{s}^B$ as $\boldsymbol{E}^A$ and $\boldsymbol{E}$, respectively. The empirical indices*
824 *are the indices $\hat{\boldsymbol{r}}$ for $(\boldsymbol{E}^A, \boldsymbol{c})$; namely, $\hat{r}_i$ is the unique solution to $\mathbf{E}_{v_i \sim E_i^A}[\max\{v_i - \hat{r}_i, 0\}] = c_i$.*
825 *The empirical counterpart of $\mathrm{DA}(\boldsymbol{F}, \boldsymbol{c})$ is $\mathrm{FPA}(\boldsymbol{E}^{\hat{r}})$. The empirical mappings are $\lambda^{\hat{\boldsymbol{r}}}$ and $\mu^{(\boldsymbol{F}, \hat{\boldsymbol{r}})}$,*
826 *computed as in Definition C.9.*

827 Note that $\mu^{(\boldsymbol{F}, \hat{\boldsymbol{r}})}$ depends on distributions while $\lambda^{\hat{\boldsymbol{r}}}$ does not. The following theorem, analogous to
828 Theorem C.11, shows that the empirical mappings $\lambda^{\hat{\boldsymbol{r}}}$ and $\mu^{(\boldsymbol{F}, \hat{\boldsymbol{r}})}$ approximately preserve the utilities
829 with high probability.

830 **Theorem C.14.** *There is $M = O\left(\frac{H^2}{\epsilon^2}\left[\log\left(\frac{H}{\epsilon}\right) + \log\left(\frac{n}{\delta}\right)\right]\right)$, such that for all $m > M$, with*
831 *probability at least $1 - \delta$ over the random draw of $\boldsymbol{s}^A$,*

832 *1. For any monotone mixed strategy profile $\boldsymbol{\beta} = (\beta_i, \boldsymbol{\beta}_{-i})$ for $\mathrm{FPA}(\boldsymbol{F}^{\hat{r}})$, for each bidder $i$,*

$$\left| u_i^{\mathrm{FPA}(\boldsymbol{F}^{\hat{r}})}(\boldsymbol{\beta}) - u_i^{\mathrm{DA}(\boldsymbol{F}, \boldsymbol{c})}(\lambda^{\hat{\boldsymbol{r}}}(\boldsymbol{\beta})) \right| \leq \epsilon.$$

833 *2. For any mixed strategy profile $\boldsymbol{\alpha} = (\alpha_i, \boldsymbol{\alpha}_{-i})$ for $\mathrm{DA}(\boldsymbol{F}, \boldsymbol{c})$, for each bidder $i$,*

$$u_i^{\mathrm{DA}(\boldsymbol{F}, \boldsymbol{c})}(\boldsymbol{\alpha}) \leq u_i^{\mathrm{FPA}(\boldsymbol{F}^{\hat{r}})}(\mu^{(\boldsymbol{F}, \hat{\boldsymbol{r}})}(\boldsymbol{\alpha})) + \epsilon.$$

834 *If $\alpha_i$ claims above $\hat{r}_i$, then we also have $u_i^{\mathrm{DA}(\boldsymbol{F}, \boldsymbol{c})}(\boldsymbol{\alpha}) \geq u_i^{\mathrm{FPA}(\boldsymbol{F}^{\hat{r}})}(\mu^{(\boldsymbol{F}, \hat{\boldsymbol{r}})}(\boldsymbol{\alpha})) - \epsilon$.*

835 Before proving Theorem C.14, we first derive a few important consequences.

836 **Corollary C.15.** *If $m > M$ as in the condition of Theorem C.14, then with probability at least $1 - \delta$,*

837 *1. For any monotone strategy profile $\boldsymbol{\beta}$, if $\boldsymbol{\beta}$ is an $\epsilon'$-NE in $\mathrm{FPA}(\boldsymbol{F}^{\hat{r}})$, then $\lambda^{\hat{\boldsymbol{r}}}(\boldsymbol{\beta})$ is an*
838 *$(\epsilon' + 2\epsilon)$-NE in $\mathrm{DA}(\boldsymbol{F}, \boldsymbol{c})$.*

839 *2. Conversely, for any strategy profile $\boldsymbol{\alpha}$ that claims above $\hat{\boldsymbol{r}}$, if $\boldsymbol{\alpha}$ is an $\epsilon'$-NE in $\mathrm{DA}(\boldsymbol{F}, \boldsymbol{c})$,*
840 *then $\mu^{(\boldsymbol{F}, \hat{\boldsymbol{r}})}(\boldsymbol{\alpha})$ is an $(\epsilon' + 2\epsilon)$-NE in $\mathrm{FPA}(\boldsymbol{F}^{\hat{r}})$.*

841 *Proof.* We prove the two items respectively,

842 1. Let $\boldsymbol{\beta} = (\beta_i, \boldsymbol{\beta}_{-i})$ be an $\epsilon'$-NE in $\mathrm{FPA}(\boldsymbol{F}^{\hat{r}})$ satisfying the condition in the statement. For
843 any strategy $\alpha_i$, by Theorem C.14 item 2,

$$u_i^{\mathrm{DA}(\boldsymbol{F}, \boldsymbol{c})}(\alpha_i, \lambda^{\hat{\boldsymbol{r}}}(\boldsymbol{\beta}_{-i})) \leq u_i^{\mathrm{FPA}(\boldsymbol{F}^{\hat{r}})}(\mu^{(\boldsymbol{F}, \hat{\boldsymbol{r}})}(\alpha_i), \mu^{(\boldsymbol{F}, \hat{\boldsymbol{r}})}(\lambda^{\hat{\boldsymbol{r}}}(\boldsymbol{\beta}_{-i}))) + \epsilon.$$

844 Since $\boldsymbol{\beta}_{-i}$ is monotone, by Claim C.10 item 2, we have $\mu^{(\boldsymbol{F}, \hat{\boldsymbol{r}})}(\lambda^{\hat{\boldsymbol{r}}}(\boldsymbol{\beta}_{-i})) = \boldsymbol{\beta}_{-i}$. Thus,

$$u_i^{\mathrm{DA}(\boldsymbol{F}, \boldsymbol{c})}(\alpha_i, \lambda^{\hat{\boldsymbol{r}}}(\boldsymbol{\beta}_{-i})) \leq u_i^{\mathrm{FPA}(\boldsymbol{F}^{\hat{r}})}(\mu^{(\boldsymbol{F}, \hat{\boldsymbol{r}})}(\alpha_i), \boldsymbol{\beta}_{-i}) + \epsilon$$

845 $\hfill \boldsymbol{\beta}$ is an $\epsilon'$-NE in $\mathrm{FPA}(\boldsymbol{F}^{\hat{r}})$

$$\leq u_i^{\mathrm{FPA}(\boldsymbol{F}^{\hat{r}})}(\boldsymbol{\beta}) + \epsilon' + \epsilon$$

846 $\hfill$ Theorem C.14 item 1

$$\leq u_i^{\mathrm{DA}(\boldsymbol{F}, \boldsymbol{c})}(\lambda^{\hat{\boldsymbol{r}}}(\boldsymbol{\beta})) + \epsilon' + 2\epsilon.$$

847     2. For any strategy $\beta_i$, by Proposition 2.2, there exists some monotone strategy $\beta_i'$, such that

$$u_i^{\mathrm{FPA}(\boldsymbol{F}^{\hat{\boldsymbol{r}}})}(\beta_i, \mu^{(\boldsymbol{F}, \hat{\boldsymbol{r}})}(\boldsymbol{\alpha}_{-i})) \le u_i^{\mathrm{FPA}(\boldsymbol{F}^{\hat{\boldsymbol{r}}})}(\beta_i', \mu^{(\boldsymbol{F}, \hat{\boldsymbol{r}})}(\boldsymbol{\alpha}_{-i})).$$

848     Then by Theorem C.14 item 1,

$$u_i^{\mathrm{FPA}(\boldsymbol{F}^{\hat{\boldsymbol{r}}})}(\beta_i', \mu^{(\boldsymbol{F}, \hat{\boldsymbol{r}})}(\boldsymbol{\alpha}_{-i})) \le u_i^{\mathrm{DA}(\boldsymbol{F}, \boldsymbol{c})}(\lambda^{\hat{\boldsymbol{r}}}(\beta_i'), \lambda^{\hat{\boldsymbol{r}}}(\mu^{(\boldsymbol{F}, \hat{\boldsymbol{r}})}(\boldsymbol{\alpha}_{-i}))) + \epsilon.$$

849     Since $\boldsymbol{\alpha}_{-i}$ claims above $\hat{\boldsymbol{r}}_{-i}$, by Claim C.10 item 1, we have $\lambda^{\hat{\boldsymbol{r}}}(\mu^{(\boldsymbol{F}, \hat{\boldsymbol{r}})}(\boldsymbol{\alpha}_{-i})) = \boldsymbol{\alpha}_{-i}$.
850 Thus

$$u_i^{\mathrm{FPA}(\boldsymbol{F}^{\hat{\boldsymbol{r}}})}(\beta_i, \mu^{(\boldsymbol{F}, \hat{\boldsymbol{r}})}(\boldsymbol{\alpha}_{-i})) \le u_i^{\mathrm{DA}(\boldsymbol{F}, \boldsymbol{c})}(\lambda^{\hat{\boldsymbol{r}}}(\beta_i'), \boldsymbol{\alpha}_{-i}) + \epsilon$$

851 $$\boldsymbol{\alpha} \text{ is an } \epsilon'\text{-NE in } \mathrm{DA}(\boldsymbol{F}, \boldsymbol{c})$$

$$\le u_i^{\mathrm{DA}(\boldsymbol{F}, \boldsymbol{c})}(\boldsymbol{\alpha}) + \epsilon' + \epsilon$$

852 $$\text{Theorem C.14 item 2}$$

$$\le u_i^{\mathrm{FPA}(\boldsymbol{F}^{\hat{\boldsymbol{r}}})}(\mu^{(\boldsymbol{F}, \hat{\boldsymbol{r}})}(\boldsymbol{\alpha})) + \epsilon' + 2\epsilon.$$

853                                                                                □

854 As a consequence of Corollary C.15 and Corollary 3.13, any approximate BNE in $\mathrm{FPA}(\boldsymbol{E}^{\hat{\boldsymbol{r}}})$ is
855 transformed by $\lambda^{\hat{\boldsymbol{r}}}$ to an approximate NE in $\mathrm{DA}(\boldsymbol{F}, \boldsymbol{c})$, as formalized by the following theorem.

856 **Theorem C.16.** *There is* $M = O\left(\frac{H^2}{\epsilon^2}\left[n \log n \log\left(\frac{H}{\epsilon}\right) + \log\left(\frac{n}{\delta}\right)\right]\right)$, *such that for all* $m > M$,
857 *with probability at least* $1 - \delta$ *over random draws of samples* $\boldsymbol{s}$, *we have: for any monotone strategy*
858 *profile* $\boldsymbol{\beta}$ *that is an* $\epsilon'$*-BNE in* $\mathrm{FPA}(\boldsymbol{E}^{\hat{\boldsymbol{r}}})$, $\lambda^{\hat{\boldsymbol{r}}}(\boldsymbol{\beta})$ *is an* $(\epsilon' + 4\epsilon)$*-NE in* $\mathrm{DA}(\boldsymbol{F}, \boldsymbol{c})$.

859 *Proof.* First use Corollary 3.13 for distributions $\boldsymbol{F}^{\hat{\boldsymbol{r}}}$. Note that $\boldsymbol{E}^{\hat{\boldsymbol{r}}}$ is an empirical product distribution
860 for $\boldsymbol{F}^{\hat{\boldsymbol{r}}}$, because $\boldsymbol{E}$ consists of samples $\boldsymbol{s}^B$, $\hat{\boldsymbol{r}}$ is determined from samples $\boldsymbol{s}^A$, and these two sets
861 of samples are disjoint. Thus, with probability at least $1 - \delta/2$ over the random draw of $\boldsymbol{s}^B$, any
862 monotone strategy profile $\boldsymbol{\beta}$ that is an $\epsilon'$-BNE in $\mathrm{FPA}(\boldsymbol{E}^{\hat{\boldsymbol{r}}})$ is an $(\epsilon' + 2\epsilon)$-BNE in $\mathrm{FPA}(\boldsymbol{F}^{\hat{\boldsymbol{r}}})$. An
863 $(\epsilon' + 2\epsilon)$-BNE must be an $(\epsilon' + 2\epsilon)$-NE in $\mathrm{FPA}(\boldsymbol{F}^{\hat{\boldsymbol{r}}})$, so by Corollary C.15, with probability at least
864 $1 - \delta/2$ over the random draw of $\boldsymbol{s}^A$, $\lambda^{\hat{\boldsymbol{r}}}(\boldsymbol{\beta})$ is an $(\epsilon' + 4\epsilon)$-NE in $\mathrm{DA}(\boldsymbol{F}, \boldsymbol{c})$.   □

865 Theorem C.16 does not include the reverse direction, i.e., from an $\epsilon'$-NE in $\mathrm{DA}(\boldsymbol{F}, \boldsymbol{c})$ to an $(\epsilon' + \epsilon)$-
866 BNE in $\mathrm{FPA}(\boldsymbol{E}^{\hat{\boldsymbol{r}}})$ (cf. Theorem C.12). This is for two reasons: (1) Such a transformation will result
867 in $(\epsilon' + 4\epsilon)$-NE in $\mathrm{FPA}(\boldsymbol{E}^{\hat{\boldsymbol{r}}})$, but $(\epsilon' + 4\epsilon)$-NE in $\mathrm{FPA}(\boldsymbol{E}^{\hat{\boldsymbol{r}}})$ is not necessarily an $(\epsilon' + 4\epsilon)$-BNE.
868 (2) Unlike interim utility, ex ante utility cannot be learned from samples directly; in other words,
869 $u_i^{\mathrm{FPA}(\boldsymbol{E}^{\hat{\boldsymbol{r}}})}(\boldsymbol{\beta})$ does not necessarily approximate $u_i^{\mathrm{FPA}(\boldsymbol{F}^{\hat{\boldsymbol{r}}})}(\boldsymbol{\beta})$ even if $\boldsymbol{\beta}$ is monotone. This is because
870 in the computation of ex ante utility we need to take expectation over bidder $i$'s own value but for
871 interim utility we do not need to take such an expectation.

872 **Proof of Theorem C.14.**   The main idea is as follows: For item 1, we need to show that the utility
873 of a strategy profile $\boldsymbol{\beta}$ in $\mathrm{FPA}(\boldsymbol{F}^{\hat{\boldsymbol{r}}})$ approximates the utility of its image $\boldsymbol{\alpha} = \lambda^{\hat{\boldsymbol{r}}}(\boldsymbol{\beta})$ in $\mathrm{DA}(\boldsymbol{F}, \boldsymbol{c})$.
874 We wish to use Theorem C.11 to do so but it cannot be used directly because $\hat{\boldsymbol{r}}$ is not the indices of
875 $(\boldsymbol{F}, \boldsymbol{c})$. Instead, we construct a set of "empirical costs" $\hat{\boldsymbol{c}}$ such that $\hat{\boldsymbol{r}}$ becomes the indices of $(\boldsymbol{F}, \hat{\boldsymbol{c}})$.
876 Then Theorem C.11 can be used to show that $u_i^{\mathrm{FPA}(\boldsymbol{F}^{\hat{\boldsymbol{r}}})}(\boldsymbol{\beta}) = u_i^{\mathrm{DA}(\boldsymbol{F}, \hat{\boldsymbol{c}})}(\boldsymbol{\alpha})$. With an additional
877 lemma (Lemma C.17) which shows that $\hat{\boldsymbol{c}}$ approximates $\boldsymbol{c}$ up to $\epsilon$-error, we are able to establish the
878 following chain of approximate equations

$$u_i^{\mathrm{FPA}(\boldsymbol{F}^{\hat{\boldsymbol{r}}})}(\boldsymbol{\beta}) = u_i^{\mathrm{DA}(\boldsymbol{F}, \hat{\boldsymbol{c}})}(\boldsymbol{\alpha}) \overset{\epsilon}{\approx} u_i^{\mathrm{DA}(\boldsymbol{F}, \boldsymbol{c})}(\boldsymbol{\alpha}).$$

879 The proof for item 2 is similar.

880 Formally, define $\hat{\boldsymbol{c}} = (\hat{c}_i)_{i \in [n]}$, where

$$\hat{c}_i \coloneqq \mathbf{E}_{v_i \sim F_i}\left[\max\{v_i - \hat{r}_i, 0\}\right]. \tag{14}$$

881 Note that $\hat{c}_i$ is determined by samples $\boldsymbol{s}^A$ since the empirical index $\hat{r}_i$ is computed from $\boldsymbol{s}^A$.

**Lemma C.17.** *There is $M = O\left(\frac{H^2}{\epsilon^2}\left[\log\frac{H}{\epsilon} + \log\frac{n}{\delta}\right]\right)$, such that if $m/2 > M$, then with probability at least $1 - \delta$ over the random draw of $\boldsymbol{s}^A$, for each $i \in [n]$, $|c_i - \hat{c}_i| \le \epsilon$.*

*Proof.* The main idea to prove this claim is to show that the class $\mathcal{H}_i = \{h^r \mid r \in [-H, H]\}$ where $h^r(x) = \max\{x - r, 0\}$ has pseudo-dimension $\mathrm{Pdim}(\mathcal{H}_i) = O(1)$ and thus uniformly converges with $O\left(\frac{H^2}{\epsilon^2}\left[\log\frac{H}{\epsilon} + \log\frac{1}{\delta}\right]\right)$ samples.

Formally, consider the pseudo-dimension $d$ of the class $\mathcal{H}_i = \{h^r \mid r \in [-H, H]\}$ where $h^r(x) := \max\{x - r, 0\}$ for $x \in [0, H]$ (thus $h^r(x) \in [0, 2H]$). We claim that $d = O(1)$. To see this, fix any $d$ samples $(x_1, x_2, \ldots, x_d)$ and any witnesses $(t_1, t_2, \ldots, t_d)$, we bound the number of distinct labelings that can be given by $\mathcal{H}_i$ to these samples. Each sample $x_j$ induces a partition of the parameter space (the space of $r$) $[-H, H]$ into two intervals $[-H, x_j]$ and $(x_j, H]$, such that for any $r \le x_j$, $h^r(x_j) = x_j - r$, and for $r > x_j$, $h^r(x_j) = 0$. All $d$ samples partition $[-H, H]$ into (at most) $d + 1$ consecutive intervals, $I_1, \ldots, I_{d+1}$, such that within each interval $I_k$, $h^r(x_j)$ is either $x_j - r$ for all $r \in I_k$ or $0$ for all $r \in I_k$, for each $j \in [d]$. We further divide each $I_k$ using witnesses $t_j$'s: for each $j \in [d]$, if $h^r(x_j) = x_j - r$ for $r \in I_k$, then we cut $I_k$ at the point $r = x_j - t_j$; in this way we cut each $I_k$ into at most $d + 1$ sub-intervals. Within each sub-interval $I' \subseteq I_k$, the labeling of the $d$ samples given by all $h^r$ ($r \in I'$) is the same. Since there are at most $(d+1)^2$ sub-intervals in total, there are at most $(d+1)^2$ distinct labelings. To pseudo-shatter $d$ samples, we must have $2^d \le (d+1)^2$, which gives $d = O(1)$.

By the definition of $\hat{r}_i$, we have

$$c_i = \mathbf{E}_{v_i \sim E_i^A}\left[\max\{v_i - \hat{r}_i, 0\}\right] = \mathbf{E}_{v_i \sim E_i^A}\left[h^{\hat{r}_i}(v_i)\right],$$

and $\hat{r}_i \in [-H, H]$. Also note that $\hat{c}_i = \mathbf{E}_{v_i \sim F_i}[h^{\hat{r}_i}(v_i)]$. Thus the conclusion $|c_i - \hat{c}_i| \le \epsilon$ follows from Theorem 3.6 and a union bound over $i \in [n]$. $\qquad\square$

**Lemma C.18.** *Suppose $|c_i - \hat{c}_i| \le \epsilon$, then for any strategies $\boldsymbol{\alpha}$,*

$$\left| u_i^{\mathrm{DA}(\boldsymbol{F}, \boldsymbol{c})}(\boldsymbol{\alpha}) - u_i^{\mathrm{DA}(\boldsymbol{F}, \hat{\boldsymbol{c}})}(\boldsymbol{\alpha}) \right| \le \epsilon.$$

*Proof.* Couple the realizations of values (and threshold prices and bids if the strategies are randomized) in $\mathrm{DA}(\boldsymbol{F}, \boldsymbol{c})$ and $\mathrm{DA}(\boldsymbol{F}, \hat{\boldsymbol{c}})$. When all bidders use the same strategies $\boldsymbol{\alpha}$ in $\mathrm{DA}(\boldsymbol{F}, \boldsymbol{c})$ and $\mathrm{DA}(\boldsymbol{F}, \hat{\boldsymbol{c}})$, bidder $i$ receives the same allocation and pays the same price (but not the same search costs) in these two auctions. The only difference in bidder $i$'s utilities is the search costs she pays, and the difference is upper-bounded by $|c_i - \hat{c}_i| \le \epsilon$. $\qquad\square$

Now we finish the proof of Theorem C.14.

*Proof of Theorem C.14.* First consider item 1. We use $a \overset{\epsilon}{\approx} b$ to denote $|a - b| \le \epsilon$. For any monotone strategies $\boldsymbol{\beta}$ for $\mathrm{FPA}(\boldsymbol{E}^{\hat{r}})$,

$$
\begin{aligned}
u_i^{\mathrm{FPA}(\boldsymbol{F}^{\hat{r}})}(\boldsymbol{\beta}) &= u_i^{\mathrm{DA}(\boldsymbol{F}, \hat{\boldsymbol{c}})}(\lambda^{\hat{r}}(\boldsymbol{\beta})) && \text{Theorem C.11 item 1}\\
&\overset{\epsilon}{\approx} u_i^{\mathrm{DA}(\boldsymbol{F}, \boldsymbol{c})}(\lambda^{\hat{r}}(\boldsymbol{\beta})) && \text{Lemma C.18.}
\end{aligned}
$$

As for item 2, for any strategies $\boldsymbol{\alpha}$ for $\mathrm{DA}(\boldsymbol{F}, \boldsymbol{c})$, by Lemma C.18,

$$u_i^{\mathrm{DA}(\boldsymbol{F}, \boldsymbol{c})}(\boldsymbol{\alpha}) \overset{\epsilon}{\approx} u_i^{\mathrm{DA}(\boldsymbol{F}, \hat{\boldsymbol{c}})}(\boldsymbol{\alpha})$$

By Theorem C.11 item 2, we have $u_i^{\mathrm{DA}(\boldsymbol{F}, \hat{\boldsymbol{c}})}(\boldsymbol{\alpha}) \le u_i^{\mathrm{FPA}(\boldsymbol{F}^{\hat{r}})}(\mu^{(\boldsymbol{F}, \hat{r})}(\boldsymbol{\alpha}))$ where "$=$" holds if $\alpha_i$ claims above $\hat{r}_i$, which concludes the proof. $\qquad\square$

## Footnotes

[2]Note that there is no private information at the beginning of the auction.

[3] We describe mappings for pure strategies here. For mixed strategies, their images are naturally distributions over the images of pure strategies under $\lambda$ and $\mu$.