[Reviews · NeurIPS 2020]

Review 1

Summary and Contributions: This paper studies the sample complexity of estimating the buyers' utilities in non-truthful auctions, such as first-price auctions and all-pay auctions, against any strategies. In this work, the samples are drawn from the prior distribution (instead of historical bids). The authors provide tight bounds of sample complexities for various auction formats, including the first-price auction, the all-pay auction, and auctions with search costs. They also give a slightly improved bound for the Pandora's Box problem. With the estimated buyers' utilities, one can compute the approximate-BNE. As an application, the authors obtain a sample-efficient and computational-efficient algorithm for computing approximate-BNE in the first-price auction.

Strengths: (1) This paper is well-written, clear, and easy to follow. (2) The authors provide full characterizations of sample complexities for various auctions, including the first-price auction, the all-pay auction, and auctions with search costs. The upper bounds and lower bounds are matched up to polylog factors. (3) This paper concerns the sample complexity of non-truthful auctions, which is relevant and interesting to the NeurIPS community, especially people working in computational economics.

Weaknesses: This paper seems to be a dense theoretical paper with many results, which does not fit the 8-page limit in NeurIPS very well (but the authors do a good job of presenting their findings and high-level intuitions.)

Correctness: The reviewer did not check the proofs in the appendix, but all the results seem sound and plausible in hindsight. The proof of Lemma 3.7 is clear and easy to follow.

Clarity: This paper is generally well written, clear, and easy to follow.

Relation to Prior Work: The authors do a good job of surveying the literature.

Reproducibility: Yes

Additional Feedback: Do the results continue to hold for any arbitrary tie-breaking rules? For example, ties are broken lexicographically. [After rebuttal: thank you for the response.]


Review 2

Summary and Contributions: The paper studies the sample complexity of learning the interim utility functions in single-item first-price auctions with bounded valuations. The main result is that with O(n / \eps^2) samples, whp the empirical utility functions on the sample set induced by all monotone bidding strategies uniformly converge to the actual interim utility functions with an additive error of \eps. Moreover, the authors give a lower bound that matches the above sample complexity up to a polylogarithmic factor. The authors further show that uniform convergence holds even if one replaces the empirical distribution with the product of the empirical marginal distributions. Combined with a recent result by Shen et al. this gives a Monte Carlo algorithm for computing \eps-BNE for first-price auctions with arbitrary (but bounded and independent across agents) valuation distributions. The authors also study extensions of their results to all-pay auctions, etc.

Strengths: The problem studied is of theoretical and practical significance. The empirical utility functions as estimators of the actual utility functions are quite natural. The main result (the sample complexity upper bound) is approximately optimal, and in particular implies an algorithm for computing \eps-BNE. The extensions also suggest the techniques are quite general and might be useful in other contexts.

Weaknesses: While I like the paper overall, given the theoretical nature of the paper, I'm a bit more inclined to evaluate it from a technical perspective, as is common at TCS venues. The authors mention that their bound on the pseudo-dimension "follows a powerful framework for bounding pseudo-dimensions, introduced by Morgenstern and Roughgarden (2016) and Balcan et al. (2018)" -- I'd appreciate it if the authors could further clarify how their proofs rely on / differ from those in previous work. Similarly, the authors say that Lemma 3.10 "is closely related to a concentration inequality by Devanur et al. (2016)" -- which makes me wonder how exactly they are related since very superficially they do look quite similar. I'm definitely not obsessed with "technical novelty" but it would be nice if the authors could make the connections clear (which I don't think would in any way diminish the contribution of the current paper).

Correctness: I believe the main results are correct, though I did not check the proofs in the appendices.

Clarity: The paper is overall well-written and carefully polished. I feel a bit lost in Section 4, where the authors discuss the model with costly search, Dutch auctions and Padora's Box in passing and present their results in similar settings, but given the rather strict length limit of NeurIPS submissions this is more or less inevitable.

Relation to Prior Work: The authors give a satisfactory discussion of related work.

Reproducibility: Yes

Additional Feedback: ===== after rebuttal ===== The response does answer my questions, and my overall evaluation remains the same. ==================== Does Theorem 3.3 hold even when the valuations are not independent?


Review 3

Summary and Contributions: This paper studies the sample complexity of learning the interim utility in a non-truthful, single-item auction, uniformly for any strategy profile, by sample access to the value distribution and ability to calculate ex post utility in the auction. The results are attained under the restriction to monotone bidding strategies. Surprisingly, this can be done in n/eps^2, for first price and all-pay auctions, and this bound is tight (they also make observations for descending price, dutch auctions). This is interesting, because by estimating interim utilities then in principle, all approximate equilibria can then be understood (since they are implied by understanding interim utility). The sample complexity bound uses pseudo-dimension.

Strengths: As best as I know, the results are novel, and are technically sophisticated even though they don't require any truly new machinery.

Weaknesses: I am not sure that the work can be immediately applied. The authors talk in the introduction about using this kind of method in the following way: (1) Evaluating the performance of a strategy given what other bidders do. --> yes, but how would an agent would form a hypothesis about what strategies others will follow? (2) Is a profile of strategies in an eps-BNE? --> yes, but how to find a candidate BNE to test? (3) Estimate the interim regret to a bidder, assuming others are truthful --> how? Just plugging in truthful strategies doesn't work. Rather, we need to find the best response to truthful strategies by others. [After rebuttal:] I appreciate the authors' responses, and suggest to find space to lift out these practical observations in the camera-ready version of the paper.

Correctness: The claims appear to be sound, with complete proofs for all results and a good understanding of the relevant literature.

Clarity: Yes, the exposition of the main results is clear.

Relation to Prior Work: Yes, this is good.

Reproducibility: Yes

Additional Feedback: The main question for me, outlined above, is exactly how this result can be applied. For example, in applying to interim regret, where does the "max over type for bidder i" get handled in making use of the methods outlined here? This needs to be handled to get at regret bounds, as per the bounds in Balcan et al. and Duetting et al. (ICML 2019)


Review 4

Summary and Contributions: This paper examines the sample complexity of learning interim utilities for monotone bidding strategies in a first price auctions. It provides two estimators to achieve this with essentially tight bounds. The first estimator introduces correlations which are undesirable for further estimation about equilibria, motivating the second estimator. An extension to auctions with costly search is sketched, with the details deferred to the supplemental material. It is also observed that the results apply to all pay auctions.

Strengths: The paper introduces a new type of question about sample complexity in auctions, namely learning interim utilities, and provides tight bounds for it in the natural starting example of first price auctions. The idea of restricting to learning for monotone strategies is natural and, while I haven’t followed the literature on sample complexity of auctions carefully enough to be fully confident about this, as far as I know novel as well. The extension to costly search refines bounds for the Pandora’s Box algorithm, which has found recent use in a number of economic settings and so may be of independent interest.

Weaknesses: The problem of learning utilities is certainly natural, but I do see a bit of a disconnect. The first paragraph starts with some very concrete motivations for why we may want to be able to estimate interim utilities. But then these motivations disappear from the rest of the paper and even the broader impact statement acknowledges that this is not intended to have an impact on practice. This seems to be a missed opportunity to me. The paper would be stronger if the sample complexity bounds derived could be connected back to showing how they can be used to inform practical decisions or recommendations in auction systems, of the sort referenced in the first paragraph. One potential issue I see with this is that the bounds apply to arbitrary choices of monotone strategies, and for most of the applications of interest some sort of assumption or information would be needed about exactly which strategies bidders are using. The extension to auctions with costly search doesn’t seem particularly well motivated beyond pointing to it as a natural problem studied in prior work. Additionally, the treatment in the main text is quite brief. So while this may be interesting from a technical perspective, it isn’t clear to me that it adds much to the paper in its current form.

Correctness: While I did not examine the supplemental material, all the key results in the paper are clearly stated and accompanied by intuition for the proof strategy or proofs for special cases which appear reasonable.

Clarity: The paper is clear and well written. The technical exposition is precise and all notation is carefully defined. The decisions about what to technical arguments to include in the main text and what to defer to the appendix seem tasteful. I have a number of small comments under additional feedback, but the only one I think is particularly significant is clarifying the issue on line 249.

Relation to Prior Work: The paper provides a good overview of related work.

Reproducibility: Yes

Additional Feedback: 10 – “connection between this setting and the first price auction”. Isn’t this setting the first price auction? 53 – If an O~ bound is tight up to a log factor, isn’t it just tight? And is log actually what is meant here or should it be polylog? 55 – The difference between this work and prior work with type samples is not explained here. In particular, the line of prior work focused on estimation for revenue maximization while this work focuses on estimating interim utilities, right? This is explained later on line 113 but it seems like it should already be mentioned here. 142 – should be the sum of the x_i, not the b_i 171 – should identify the utility learning algorithm as A 175 – I would say “take B” rather than “restrict B” here because a particular choice of B is assumed. 249 – I’m a bit unclear about the issue of correlation being discussed here. F is assumed to be a product distribution, so the s_j^i aren’t correlated (which was my first reading of what was being claimed here). Looking at the way Empp is designed I can see how this breaks apart the original samples exploiting that we have a product distribution, but I’m still not clear exactly what the crucial correlations being broken are (although I haven’t studied the proof of Corollary 3.13 to try and figure this out). Post response comments: To clarify my comments on the correlation, I did eventually figure out that the correlations were what you meant, but I'm still not clear on why breaking them is important for your results.

[Author Response · NeurIPS 2020]

We thank the reviewers for their careful reviews! Below we restate the questions in blue.

**Review 1.** Do the results continue to hold for any arbitrary tie-breaking rules? For example, ties are broken
lexicographically. Our results hold for all tie-breaking rules by which any bidder $i$'s allocation is deterministically
determined by the comparisons of her bid $b_i$ with the other bidders' bids, and not affected by any other information
(such as the specific value of $b_i$). More formally, the allocation for bidder $i$ should be determined by a vector $\boldsymbol{y}_i \in \{<$
$, =, >\}^{n-1}$, which records the comparison between $b_i$ and every other bidder's bid. Breaking ties lexicographically
satisfies these conditions. We remark that ties occur with probability 0 in equilibrium; we discuss tie-breaking only for
the sake of utility learning.

**Review 2.** Clarify how the proofs rely on / differ from those in previous work. Morgenstern and Roughgarden (2016)
and Balcan et al. (2018) proposed the following framework for bounding the psuedodimension of a class $\mathcal{F}$ of functions:
given samples that are to be shattered and for any (fixed) witnesses, one classifies the functions in $\mathcal{F}$ into categories, so
that functions in the same category must output the same label on all the samples; by counting and bounding the number
of such categories, one can bound the number of shattered samples. Our proof follows this strategy. To bound the
number of categories, the argument makes use of monotonicity of bidding functions, which is specific to our problem.

Relationship between Lemma 3.10 and existing work. Devanur et al. (2016) show the concentration of the expectation
of a single function on the empirical product distribution. Lemma 3.10 of our paper generalizes this so that the
concentration applies simultaneously to a family of functions. This is more convenient for proving uniform convergence.
Our proof follows arguments similar to those of Devanur et al.

Does Theorem 3.3 hold even when the valuations are not independent? The answer is no. When values are correlated, a
bidder $i$'s value $v_i$ gives information on the other bidders' values, and so interim utilities must be defined with respect
to *conditional* distributions (over $\boldsymbol{v}_{-i}$) that vary for each $v_i$; this changes the problem substantially. In general, utilities
over each conditional distribution require $\Omega(n/\epsilon^2)$ samples to learn; in total, $\Omega(|T_i|n/\epsilon^2)$ samples are needed.

**Review 3.** (1) How would an agent form a hypothesis about what strategies others will follow? In modern ad auctions
(such as oCPA/oCPA ads), bidding decisions are often handed over to the platform, which also estimates the utilities for
the bidders. The platform is the entity that is most likely to have access to both value samples and bidding strategies for
such estimates.

(2) How to find a candidate BNE to test? One of the consequences of this work is that, with $\tilde{O}(n/\epsilon^2)$ samples, one
can not only test whether a strategy profile is at approximate equilibrium, but also compute approximate equilibria
(see Corollary 3.14). In particular, one may simply compute an approximate equilibrium on the empirical product
distribution on the samples, using an existing algorithm (e.g., Shen et al. 2020). The strategies form an approximate
equilibrium on the original distribution as well (see Corollary 3.13).

(3) How [does one estimate the interim regret to a bidder]? As a consequence of our results, it suffices to compute
a bidder $i$'s interim regret assuming others' values are drawn from the empirical distribution. Balcan et al. (EC'
19) compute the regret by discretizing bidder $i$'s type space $[0, H]$ into $1 + H/\epsilon$ points, $P = \{0, \epsilon, 2\epsilon, \ldots, H\}$, and
computing $\max_{v_i, b_i \in P}\{\mathbf{E}_{\boldsymbol{v}_{-i} \sim \text{emp}}[U_i(v_i, b_i, \boldsymbol{v}_{-i}) - U_i(v_i, v_i, \boldsymbol{v}_{-i})]\}$. This estimate is within $O(\epsilon)$ the actual interim
regret ($O(\epsilon)$ discretization error + $O(\epsilon)$ sampling error). In a first price auction, the best response on the empirical
distribution can be computed even without discretization. A key observation is that a best-responding bid must equal a
value that appears in a sample of an opponent's value — any other bid can be lowered to decrease payment without
affecting the allocation. One may therefore enumerate the $m(n-1)$ values to search for a best-responding bid.

**Review 4.** How to connect the sample complexity bounds derived back to showing how they can be used to inform
practical decisions or recommendations in auction systems. Please see our responses to Reviewer 3's questions.

10 - "Connection between this setting and the first price auction." "This setting" refers to auctions with search costs.

55 - ...the line of prior work focused on estimation for revenue maximization while this work focuses on estimating
interim utilities, right? Yes, we will clarify this.

249 - Correlation of the empirical distribution. Even though $\boldsymbol{F}$ is a product distribution, the empirical distribution, which
is the uniform distribution over samples $\{\boldsymbol{s}^1, \ldots, \boldsymbol{s}^m\}$, is correlated. For example, given two samples $\boldsymbol{s}^1 = (1, 3)$ and
$\boldsymbol{s}^2 = (2, 4)$, the empirical distribution is the uniform distribution over $\{(1, 3), (2, 4)\}$, which is correlated. $\mathbb{E}$mpp in this
example estimates utility on the uniform distribution over $\{(1, 3), (1, 4), (2, 3), (2, 4)\}$, which is a product distribution.

142, 171, 175 - We agree with your comments and will make the suggested changes. Thanks!

[Meta-Review · NeurIPS 2020]

This paper has been rated "Good paper - Accept" by the 4 expert reviewers and the discussion was obviously pretty quick. It is definitely an interesting and good paper. Happy to recommend acceptance as well